# Wake Development in Floating Wind Turbines: New Insights and Open Dataset from Wind Tunnel Experiments

Alessandro Fontanella[1], Alberto Fusetti[2], Stefano Cioni[3], Francesco Papi[3], Sara Muggiasca[1], Giacomo Persico[2], Vincenzo Dossena[2], Alessandro Bianchini[3], and Marco Belloli[1]

[1]Department of Mechanical Engineering, Politecnico di Milano, via La Masa 1, 20156 Milano, Italy.
[2]Department of Energy, Politecnico di Milano, Via Lambruschini 4, 20156 Milano, Italy.
[3]Department of Industrial Engineering, Università degli Studi di Firenze, Via di Santa Marta 3, 50139 Firenze, Italy.

**Correspondence:** Alessandro Fontanella (alessandro.fontanella@polimi.it)

**Abstract.** Floating offshore wind turbines represent a promising advancement in renewable energy, yet they remain in early development stages with limited performance data. As part of the NETTUNO research project, this study investigates how platform motion affects the aerodynamics of a floating wind turbine rotor and connects its load response to the wake evolution. Wind tunnel experiments were conducted using a 1:75 scale wind turbine model subjected to platform motions in various directions. Measurements include rotor loads and wake velocities at downstream distances from 3 to 5 rotor diameters. The results show that surge and pitch motions induce periodic thrust fluctuations, leading to significant variations in near-wake velocity, that have maximum amplitude at a reduced frequency of 0.6. Yaw motion causes oscillations in yaw moment and lateral wake meandering, while combined surge and sway motions result in skewed apparent wind, causing both wake velocity fluctuations and lateral meandering. Increased turbulence intensity near the wake center suggests enhanced mixing and potentially faster wake recovery beyond 5 rotor diameters, which is the furthest distance examined in the experiment. This new experimental dataset serves as a foundation for validating numerical simulation tools and provides valuable insights for optimizing the design and layout of future large-scale floating wind farms.

## 1 Introduction

Floating wind power is the technology that can open the doors of deep-water sites to wind energy projects. At present, there is only a limited number of pilot floating wind plants with project sizes ranging from 10 to 50 MW (Barter et al., 2020). However, upcoming projects are significantly larger and will include from tens to hundreds of wind turbines, reaching gigawatt scale. On that scale, wind turbines in the farm are going to interact through wakes–areas behind rotors with lower energy and higher turbulence compared to free-stream wind (Meyers et al., 2022). To date, research on floating wind power has mostly focused on individual turbines, but upcoming projects raise concerns about wakes and aerodynamic interactions inside the wind farms. Unlike bottom-fixed turbines, floating turbines may experience significant movements that impact rotor aerodynamics and wake formation. Moreover, full-scale measurements of floating wind turbine wakes are scarce. Very recently, Özinan et al. (2024) studied the near wake of a 2 MW floating wind turbine and found no evident impact of wave-induced motions on the average velocity of the wake, partially contrasting theoretical expectations. The authors of the study acknowledge that the

conclusions are uncertain due to limitations in estimating wind speed, turbulence intensity, wind direction, and the lack of information about atmospheric stability and floater motions. In this context, controlled wind tunnel experiments are essential to interpret full-scale measurements and support their conclusions.

To date, wind tunnel experiments on the effects of platform motion on the wake have been conducted using model turbines and porous disks. Bayati et al. (2017a) examined the effect of surge motion on the near wake of a 1:75 scale model of the DTU 10-MW wind turbine of Bak et al. (2013) and discovered that the wake has wind speed fluctuations due to motion. Fontanella et al. (2021) expanded the experiment to include more surge motion conditions and described how motion influences the formation of tip vortices near the rotor. Fontanella et al. (2022a) carried out an experiment with a 1:100 scale model of the IEA 15 MW turbine of Gaertner et al. (2020), which was subjected to platform movements in different directions. Their study concluded that the average velocity and turbulence in the near wake were marginally lower with motion compared to the fixed case. Messmer et al. (2024a) experimentally investigated the wake of a floating wind turbine experiencing harmonic sway and surge movements in a wind tunnel with laminar inflow conditions. They found that sway movements cause wake meandering, while surge movements create a pulsing wake, and both types of motion improve wake mixing. Messmer et al. (2024b) studied in the wind tunnel the wake of a scale model wind turbine subjected to surge motion with inflow turbulence intensity up to 3%. Their research indicates that increased turbulence in the incoming wind reduces the influence of platform motion on the wake recovery.

Porous disks have been utilized to investigate the wakes of floating wind turbines within atmospheric boundary layer flows. Schliffke et al. (2020) found that at 4.6 rotor diameters downstream, surge motion does not alter wake mean velocity, but it does affect turbulence intensity and turbulent kinetic energy profiles. Schliffke et al. (2024) demonstrated that harmonic platform motion produces distinct frequency signatures in the far wake spectra, whereas broadband motion leaves no easily discernible marks.

The effects of platform motions in different directions on the wake of a floating wind turbine remain uncertain, and the interactions among floating wind turbines through their wakes have yet to be studied. The NETTUNO project (Nettuno, 2023) aims to address these gaps by combining wind tunnel experiments with multi-fidelity simulations. In its initial phase, the project investigates how large platform movements affect wake development, while the subsequent phase will focus on wake-induced interactions between two floating wind turbines. This article presents a unique experimental approach with two main objectives. First, it examines how platform movements—typical of floating wind turbines and occurring in various directions—impact rotor aerodynamic loads. Second, it links the rotor response to wake development. Unlike previous studies on floating wind turbine wakes, this work provides a publicly available dataset that includes extensive wake measurements at multiple downstream locations under various platform motion conditions. Furthermore, the wind tunnel experiment is conducted on a larger rotor, extensively studied in past international projects, offering robust insights into wake evolution in floating wind turbines. This study addresses two key research questions:

– How do platform movements in different directions impact rotor aerodynamics and wake dynamics?

- How do variations in movement frequency and amplitude affect wake recovery and turbulence at various distances from the rotor?

To analyze these questions, we propose the following hypotheses:

- Platform motion-induced fluctuations in thrust force significantly alter wake structure and turbulence.

- The wake response closely follows the periodicity of platform motion.

Insights into wake evolution from the experiment can be utilized to optimize wind farm layouts, design floaters, and enhance control strategies at both the turbine and farm levels. Collected measurement data are accessible to the community and serve as a foundation for validating numerical simulation tools.

The paper is organized as follows. Section 2 presents the setup used for carrying out the wind tunnel experiment, which includes the wind turbine scale model and the measurements taken in the test campaign. Section 3 describes the test scenarios. Section 4 explains the processing that was applied to measurement data collected in the experiment. Key results are discussed in Sect. 5. Section 6 draws the conclusions and proposes suggestions for future work.

## 2 Wind tunnel experimental setup

The experimental campaign was conducted in the atmospheric-boundary layer test section of Politecnico di Milano wind tunnel, which is $13.84\,\mathrm{m}$ wide by $3.84\,\mathrm{m}$ high by $35\,\mathrm{m}$ long. The test turbine, shown in Fig. 1, has a rotor diameter ($D$) of $2.38\,\mathrm{m}$ and it is mounted on a six-degrees-of-freedom robotic platform that mimics the rigid-body motions of floating foundations. This setup has been used in various experiments about the aerodynamics of floating wind turbines and the data from this turbine, collected in earlier projects (Fontanella et al., 2021), were recently utilized in the International Energy Agency Task 30 to verify the accuracy of the aerodynamic response predicted by various offshore wind modeling tools (Bergua et al., 2023; Cioni et al., 2023).

### 2.1 Wind turbine

The wind turbine is a 1:75 scaled version of the DTU 10-MW (Bak et al., 2013); its geometry is summarized in Table 1. The design of the rotor blades was aimed at mimicking the load distribution of the DTU 10-MW blades, achieving similar thrust and power coefficients. To meet this goal, the wind speed was adjusted by a scale of 1:3 while ensuring the tip-speed-ratio remained the same as that of the full-scale turbine. Due to the reduced wind speed, the wind turbine scale model has lower chord-based Reynolds numbers ($50 \cdot 10^4$ to $1.5 \cdot 10^5$) compared to those of the full-scale turbine ($10^6$ to $10^7$). Since the Reynolds number in the wind tunnel is approximately 200 times lower than that of the full-scale wind turbine, the rotor of the scale model was designed using the low-Reynolds-number airfoil SD7032, which offers favorable aerodynamic characteristics: minimal sensitivity of lift to Reynolds number for angles of attack below 10 degrees, a linear lift curve, and no nonlinearities in drag (Fontanella et al., 2021). Based on this airfoil choice, the blade chord and twist distributions were modified relative to those of

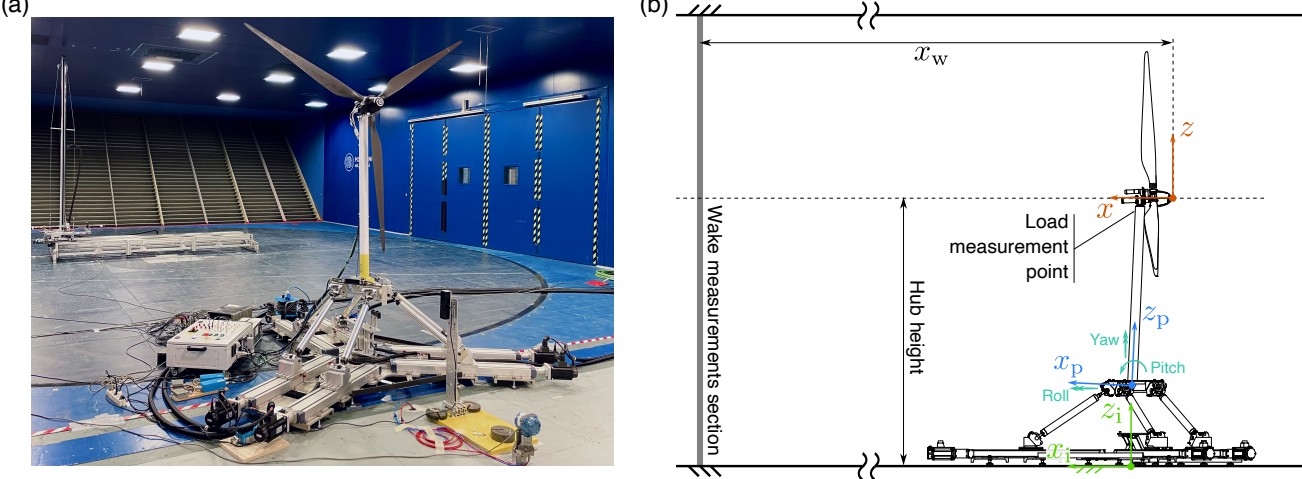

**Figure 1. (a)**: the wind turbine scale model on top of the robotic platform during testing. The traversing system used to measure the wake is visible in the background. **(b)**: schematic representation of the experimental setup with the coordinate systems: $(x_i - y_i - z_i)$ is the inertial reference frame, $(x_p - y_p - z_p)$ is the platform reference frame, and $(x - y - z)$ is the hub reference frame. $x_w$ is the distance from the wind turbine hub to the section where wake measurements were conducted.

the DTU 10-MW reference turbine to match the thrust force of the full-scale system at scale (Bayati et al., 2017b). Thrust was prioritized in the design because it is tightly coupled to the rigid-body motion of floating wind turbines and directly influences the wake velocity deficit. This scaling approach ensures that the wake generated by the scale model accurately replicates that of a full-scale turbine (Wang et al., 2021). The aerodynamic performance of the scale rotor was validated prior to this experiment, with results detailed in Appendix A. The scale rotor effectively replicates the thrust of the full-scale turbine, though it produces slightly less torque.

The structural design of the blades and tower focused on maximizing stiffness to minimize their aeroelastic response. This design philosophy aims to distinctly differentiate the wind turbine aerodynamic loads and wake response due to platform motion from the response caused by the flexibility of blades and tower, which is not within the scope of this study.

The shaft of the wind turbine scale model had a tilt of $5°$, matching the full-scale version of the DTU 10-MW. During wind tunnel tests, the tower was tilted at a negative angle of $5°$ to align the wind turbine shaft with the mean flow and ensure the rotor was vertical to the wind tunnel floor. This simplification aimed to limit loads to the axial thrust and moment in scenarios without prescribed platform motion.

## 2.2 Measurements

The test campaign measured the three forces and three moments at the interface between tower-top and nacelle, the undisturbed wind speed, and the streamwise velocity in the turbine wake. Rotor forces were measured using a six-component force

**Table 1.** Geometry of the wind turbine used in the experiment.

| Parameter | Unit | Value |
|---|---|---|
| Rotor diameter ($D$) | m | 2.381 |
| Blade length | m | 1.102 |
| Hub diameter | m | 0.178 |
| Hub height | m | 2.190 |
| Rotor overhang | m | 0.139 |
| Shaft tilt angle | ° | 5 |
| Tower-to-shaft distance | m | 0.064 |
| Tower length | m | 1.400 |
| Tower diameter | m | 0.075 |
| Tower base offset | m | 0.730 |

transducer (ATI Mini45 SI-580-20) mounted at the top of the tower. Movements of the robotic platform were recorded with MEL M5L/200 laser sensors and the rotor rotational speed with a magnetic encoder.

The undisturbed wind speed was measured at two locations using Pitot tubes. One tube was positioned 7.5 m ahead of the wind turbine at hub height, while the second was placed laterally at $x_i = 2$ m, $y_i = 5$ m, and $z_i = 3$ m. The Pitot tube positioned upstream of the wind turbine recorded the free-stream wind velocity, while the lateral Pitot tube measured the wind speed affected by blockage effects.

The streamwise velocity in the turbine wake was measured using a hot wire anemometer. A DANTEC 55P11 single-sensor
probe with a 5 $\mu$m wire diameter was employed. The wire is perpendicular to the probe axis, and it is capable of measuring mean and fluctuating velocities of one-dimensional flow. In the wind tunnel campaign, the velocity in the $x_i$ direction ($u$) was measured, consistent with other recent experiments (Messmer et al., 2024a,b). While certain motion conditions in the experiment may induce oscillations in the crosswind components, their time-averaged values are expected to be an order of magnitude lower than $u$ in the studied wake region. The probe was connected to DISA55M systems operating in constant
temperature mode, providing a high dynamic response of up to tens of kHz, making it ideal for turbulence measurements. Prior to testing, the probe was calibrated using a reference nozzle over the expected velocity range.

The use of a single hot-wire anemometer in this study was based on the assumption that the wake response to platform motion is predominantly periodic. Spatial correlation of velocity measurements was reconstructed by phase-aligning the data with the platform motion signal. To describe the spatial correlation of turbulence components that are different from the flow
structures associated with the platform periodic motion, simultaneous measurements at multiple points are necessary. This could be achieved using arrays of hot-wire probes or advanced velocimetry techniques such as Particle Image Velocimetry (PIV), which allow for detailed and simultaneous velocity field acquisition.

The measurement uncertainty for the hot wire system was determined using a Monte Carlo simulation, with uncertainties taken into account throughout the entire measurement process, beginning with the probe calibration. Factors contributing to the uncertainty included the accuracy of the data acquisition board and the pressure transducer, which influences both the reference flow velocity during calibration and the upstream wind speed during wind tunnel tests. Additionally, the uncertainty of the thermocouple, which adjusts the hot wire voltage for ambient temperature variations, was considered. The resulting average extended uncertainties, at a 95% confidence level, are $0.17 \, \mathrm{m \, s^{-1}}$ for flow speed and 0.4% for turbulence intensity.

The hot wire probe was mounted on an automatic traversing system with two motorized axes, allowing movement in the crosswind $(x_i - y_i - z_i)$ plane. Velocities were measured along lines centered on the rotor hub, both in the horizontal direction (along the $y_i$-axis) and in the vertical direction (along the $z_i$-axis). Horizontal measurements were made at 35 evenly spaced points, and vertical measurements at 27 points. The distance between consecutive measurement point is 0.1 m. The traversing system was positioned at three distances downstream of the rotor to study wake evolution: $x_w = [3, 4, 5]D$.

## 3 Test scenarios

The test scenarios were chosen by revisiting previous studies and considering recent findings in floating wind turbine aerodynamics. While some turbine operating parameters and motion conditions were kept consistent with past research for continuity, other motion cases were selected to induce various excitations in the turbine wake. All tests were conducted in low-turbulence wind conditions, with relatively uniform velocity and turbulence intensity across the wind tunnel section. This choice was made to isolate the effects of platform motion on rotor aerodynamics and wake dynamics. A low-turbulence inflow provides a controlled and repeatable environment, allowing for a more precise evaluation of the wake structure and the turbulence generated directly by the turbine and platform motion. However, we recognize that offshore environments typically exhibit moderate turbulence intensities (e.g., 3–6%), shaped by atmospheric boundary layer dynamics and wave conditions (Türk and Emeis, 2010). As a result, the wake behavior observed in this study may differ from that in real-world conditions. At the same time, large-eddy simulations of Pagamonci et al. (2025) have shown that the experiment low-turbulence inflow is not equivalent to a laminar inflow and still has a significant impact on wake development.

Prior to testing, wind characteristics were measured and are shown in Fig. 2. The mean velocity over the rotor area varies by 5% compared to the hub mean velocity and the average turbulence intensity across the rotor disk is 1.5%, varying from 1.2% to 2.1%. These spatial variations of mean velocity and turbulence intensity are consistent with those reported in other wind tunnel facilities conducting experiments on wind turbines (Schottler et al., 2018). The spatial variability in mean wind speed and turbulence intensity originates from the wind tunnel configuration. The facility has 14 fans arranged in two rows to generate airflow, which introduces inflow inhomogeneity within the return duct that houses the atmospheric-boundary layer test section. Additionally, turbulence intensity increases near the wind tunnel ceiling due to interactions between the airflow and the tunnel walls. Figure 2c displays the power spectral density of wind speed at the wind turbine hub. The air density during testing was $1.185 \, \mathrm{kg \, m^{-3}}$.

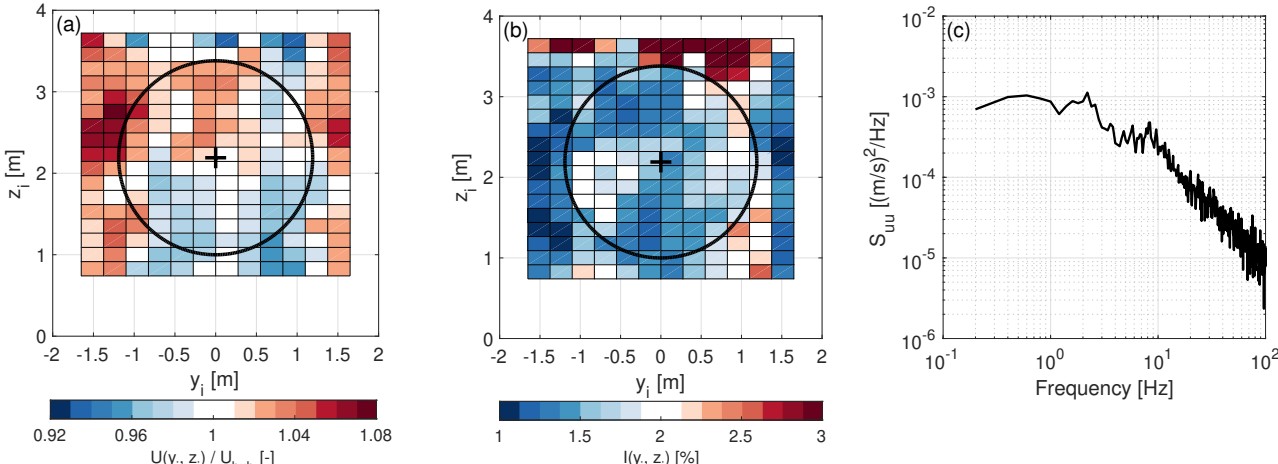

**Figure 2.** Wind speed at the wind turbine location measured prior to testing. **(a, b)** Normalized mean velocity and turbulence intensity ($I$) in the wind tunnel test section; the black circle marks the rotor edge. **(c)** Wind velocity power spectral density at the wind turbine hub.

Every test scenario is defined by the free-stream average wind velocity $U_0$, the wind turbine rotor speed, the amplitude of platform motion ($a$), the motion frequency ($f$), and the oscillation amplitude of the apparent wind speed due to platform displacement ($\Delta u$). The motion frequency relates to the free-stream wind through the rotor reduced frequency:

$$f_\mathrm{r} = \frac{fD}{U_0}. \tag{1}$$

The reduced frequency is a normalized frequency comparable to the Strouhal number, which is commonly utilized in studies of unsteady fluid dynamics.

### 3.1 Wind conditions and wind turbine settings

The wind turbine was tested at the free-stream wind speed of $4\,\mathrm{m\,s^{-1}}$ that was also investigated by Fontanella et al. (2021). The wind turbine was operated with a constant rotor speed of 240 rpm, corresponding to the optimal tip-speed ratio of 7.5, and the blade pitch was fixed to $0°$ (i.e., the wind turbine optimal pitch angle). The rotor speed and blade pitch were set to a constant value to avoid any controller actions that could affect the aerodynamic response. Reproducing the actions of a wind turbine feedback controller in scale model experiments is challenging and introduces uncertainty in the measurement of aerodynamic loads (Fontanella et al., 2023). The blade pitch was fixed individually for each blade using an inclinometer and the procedure was repeated regularly throughout the test campaign. The resulting zero blade-pitch position achieved an accuracy of $\pm 0.75°$.

The wind turbine creates about an 8% geometrical blockage effect, calculated from the rotor area and the wind tunnel test section ratio. This causes the wind speed to rise at the rotor section depending on the turbine operating conditions. The wind speed at the rotor, affected by blockage effects, was measured using the lateral Pitot tube and was $4.2\,\mathrm{m\,s^{-1}}$ in all cases. Blockage effects, which accelerate the flow around the scale model and increase the effective inflow velocity, lead to an overes-

timation of thrust and torque compared to a wind turbine operating in unconfined atmospheric conditions. A velocity increase of $0.2\,\mathrm{m\,s^{-1}}$ due to blockage over an undisturbed wind speed of $4\,\mathrm{m\,s^{-1}}$ results in aerodynamic loads that are approximately 10% higher than those in an unconfined flow, assuming identical rotor performance coefficients. This increase in loads has been thoroughly investigated by Bergua et al. (2023) by comparing numerical codes of varying fidelity to an experiment that used the same turbine as in the present work. The study demonstrated that blockage does not alter the fundamental physics of the rotor response; its effect on aerodynamic loads is predictable and can be corrected by accounting for the velocity increase.

## 3.2 Platform motion conditions

Platform motions were defined using the inertial reference frame $(x_i - y_i - z_i)$ and the platform reference frame $(x_p - y_p - z_p)$ that are depicted in Figure 1b. The inertial reference frame has the origin $O_i$ on the wind tunnel floor in correspondence of the tower axis when the robotic platform is at rest; $z_i$ points upward, $x_i$ points in the downwind direction and $y_i$ forms a right-hand backhoe with $x_i$ and $z_i$. The platform reference frame is fixed to the robotic platform and moves with the turbine; it has the origin $O_p$ in correspondence of the tower base, on the tower centerline; $z_p$ points along the tower axis, $x_p$ points in the downwind direction when the platform is at rest and $y_p$ forms a right-hand backhoe with $x_p$ and $z_p$. When the wind turbine is in its rest position, the vertical distance between $O_p$ and $O_i$ is $0.73\,\mathrm{m}$.

The system was studied under harmonic motion of different frequencies and amplitudes. The platform displacement, whether it involves a translation or a rotation, is:

$$d(t) = a \cdot \sin\left(2\pi f t\right). \tag{2}$$

The platform velocity is:

$$\dot{d}(t) = a \cdot 2\pi f \cdot \sin\left(2\pi f t\right). \tag{3}$$

The wind turbine was forced to oscillate in different directions resulting in several motion scenarios that are summarized in Table 2. The surge motion corresponds to the translation of $O_p$ along $x_i$. Pitch is the rotation around the $y_i$ axis, roll is the rotation around the $x_i$ axis, and yaw the rotation around the $z_i$ axis; the rotations are illustrated in Fig. 1. Furthermore, the experiment explored platform movements at different angles relative to the wind direction. The surge–sway motion at an angle $\gamma$ is the combination of a translation $d(t)\cos\gamma$ in the $x_i$ direction and translation $d(t)\sin\gamma$ in the $y_i$. The rotation of the platform at an angle $\gamma$ with respect to the wind direction is defined as the combination of a roll rotation around the $x_i$ axis, $d(t)\cos\gamma$, and a pitch rotation, $d(t)\sin\gamma$ around the $y_i$ axis.

The amplitudes of surge, pitch, surge-sway, and roll-pitch motions were selected to achieve values of $\Delta u$ between $0.1\,\mathrm{m\,s^{-1}}$ and $0.6\,\mathrm{m\,s^{-1}}$. When combined with a free-stream wind speed of $4\,\mathrm{m\,s^{-1}}$, these result in apparent wind-to-undisturbed wind speed ratios ($\Delta u/U_0$) ranging from 2.5% to 15%. Comparable values were taken into account in previous wind tunnel studies on the aerodynamic rotor loads of floating wind turbines (Schulz et al., 2024). The yaw motion amplitude was limited to a maximum rotation of 3° by the robotic platform. The motion frequencies were selected to replicate the reduced frequencies investigated in previous experimental studies on floating wind turbine aerodynamics, facilitating direct comparisons with existing results (Schulz et al., 2024; Bergua et al., 2023). Additionally, these frequencies are representative of the behavior of

**Table 2.** Test scenarios and wake measurements taken for each of them. Loads were measured in all cases. "H" stands for horizontal and "V" for vertical.

| Direction of Platform motion | Amplitude ($a$) [m] or [°] | Frequency ($f$) [Hz] | Reduced frequency ($f_r$) [-] | $\Delta u$ [m/s] | Measurement section | Measurement direction |
|---|---|---|---|---|---|---|
| Fixed | – | – | – | – | $3D, 4D, 5D$ | H, V |
| Surge | 0.032 m | 0.5 | 0.3 | 0.1 | $3D$ | H, V |
| | 0.064 m | 0.5 | 0.3 | 0.2 | $3D$ | H |
| | 0.016 m | 1.0 | 0.6 | 0.1 | $3D$ | H |
| | 0.032 m | 1.0 | 0.6 | 0.2 | $3D, 4D, 5D$ | H, V |
| | 0.048 m | 1.0 | 0.6 | 0.3 | $3D$ | H |
| | 0.016 m | 2.0 | 1.2 | 0.2 | $3D$ | H |
| | 0.032 m | 2.0 | 1.2 | 0.4 | $3D$ | H ,V |
| | 0.048 m | 2.0 | 1.2 | 0.6 | $3D$ | H |
| Pitch | 1.3° | 0.5 | 0.3 | 0.1 | $3D, 5D$ | H, V |
| | 2.5° | 0.5 | 0.3 | 0.2 | $3D$ | H, V |
| | 3.0° | 0.5 | 0.3 | 0.25 | $3D$ | H, V |
| | 0.6° | 1.0 | 0.6 | 0.1 | $3D$ | H, V |
| | 1.3° | 1.0 | 0.6 | 0.2 | $3D, 4D, 5D$ | H, V |
| | 1.9° | 1.0 | 0.6 | 0.3 | $3D$ | H, V |
| | 0.3° | 2.0 | 1.2 | 0.1 | $3D$ | H, V |
| | 0.6° | 2.0 | 1.2 | 0.2 | $3D$ | H, V |
| | 1.3° | 2.0 | 1.2 | 0.4 | $3D, 5D$ | H, V |
| | 1.9° | 2.0 | 1.2 | 0.6 | $3D$ | H, V |
| Surge–sway 30° | 0.032 m | 0.5 | 0.3 | 0.1 | $3D$ | H |
| Surge–sway 15° | 0.032 m | 1.0 | 0.6 | 0.19 | $3D$ | H |
| Surge–sway 30° | 0.032 m | 1.0 | 0.6 | 0.17 | $3D, 5D$ | H |
| Surge–sway 45° | 0.032 m | 1.0 | 0.6 | 0.14 | $3D$ | H |
| Surge–sway 30° | 0.032 m | 2.0 | 1.2 | 0.35 | $3D$ | H |
| Roll–pitch 15° | 1.3° | 1.0 | 0.6 | 0.20 | $3D$ | H |
| Roll–pitch 30° | 1.3° | 1.0 | 0.6 | 0.18 | $3D$ | H |
| Roll–pitch 45° | 1.3° | 1.0 | 0.6 | 0.15 | $3D$ | H |
| Yaw | 2° | 0.5 | 0.3 | – | $3D$ | H |
| | 2° | 1.0 | 0.6 | – | $3D, 5D$ | H |
| | 3° | 1.0 | 0.6 | – | $3D$ | H |
| | 2° | 2.0 | 1.2 | – | $3D$ | H |

full-scale floating wind turbines (Wise and Bachynski, 2020): reduced frequencies of 0.3 and 0.6 correspond to full-scale frequencies of 0.02 Hz and 0.04 Hz, typical of rigid-body modes in floating turbines; the reduced frequency of 1.2, on the other hand, corresponds to a full-scale frequency of 0.08 Hz, representative of motions at wave frequency.

The wind turbine experiences an apparent wind when the system undergoes platform motion. The apparent wind velocity results from combining the incoming wind velocity ($U_0$) with the rotor structural velocity, as shown in Fig. 3.

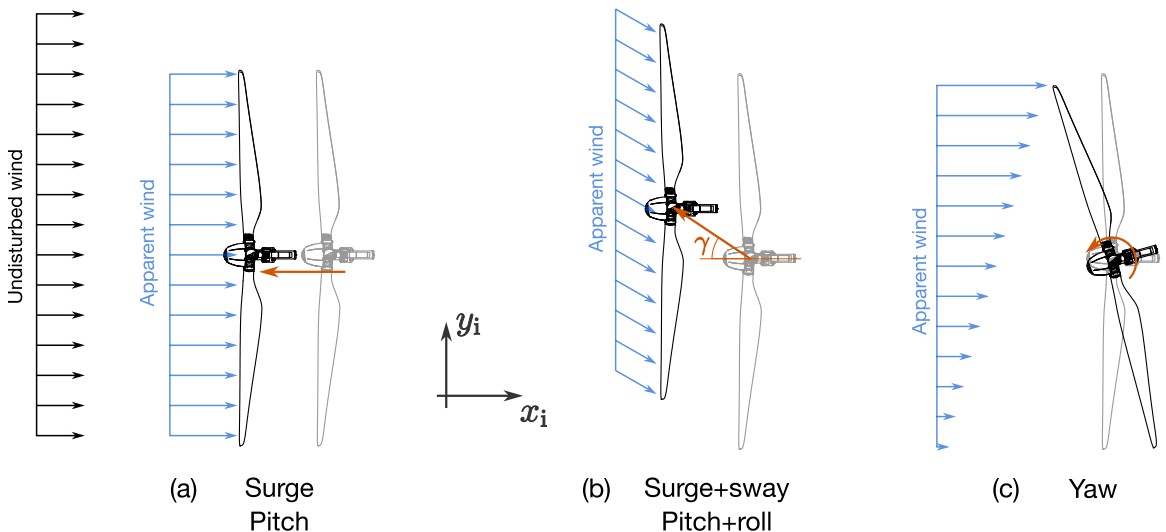

**Figure 3.** The motion conditions studied in the experiment are classified into three categories: **(a)** surge and pitch motions that cause an apparent wind velocity that is aligned to the undisturbed wind; **(b)** motions that combine surge with sway, or pitch with roll, resulting in an apparent wind velocity with a crosswind component; **(c)** yaw movement, resulting in an apparent wind velocity gradient across the rotor span.

With surge and pitch motions the rotor is forced to oscillate in the $x_i$ direction against the incoming wind. With surge, the apparent wind speed is:

$$u_a(t) = U_0 - \dot{d}(t), \tag{4}$$

The oscillation amplitude of the apparent wind speed is $\Delta u = a \cdot 2\pi f$.

The pitch motion results in a skewed flow due to the rotor plane tilt angle. However, the pitch amplitude is relatively small, thus the rotor motion is primarily in the $x_i$ direction and can be approximated with the hub motion. The apparent wind speed is:

$$u_a(t) = U_0 - \dot{d}_{\text{hub},x}(t), \tag{5}$$

with $\dot{d}_{\text{hub},x} = \dot{d}(t) \cdot r_{\text{hub}}$, where $r_{\text{hub}}$ is the hub distance between $O_p$ and the rotor apex.

When surge and sway or pitch and roll movements combine, they generate a flow with considerable skew. The apparent wind has velocity components in the $x_i$ and $y_i$ directions. The component in the $x_i$ direction is obtained using Eq. 5, setting $\dot{d}_{\text{hub},x}(t) = \dot{d}(t) \cdot \cos\gamma$ for the surge–sway case and $\dot{d}_{\text{hub},x} = \dot{d}(t) \cdot \cos\gamma \cdot r_{\text{hub}}$ for the roll–pitch case. The velocity component in the $y_i$ direction is:

$$v_a(t) = -\dot{d}_{\text{hub},y}(t), \tag{6}$$

where $\dot{d}_{\mathrm{hub},y} = \dot{d}(t) \cdot \sin\gamma$ for the surge–sway case and $\dot{d}_{\mathrm{hub},y} = \dot{d}(t) \cdot \sin\gamma \cdot r_{\mathrm{hub}}$ for the roll–pitch case.

During yaw motion, because the yaw amplitude is small, the apparent velocity is mainly in the $x_{\mathrm{i}}$ direction. It has a gradient across the rotor width, and its profile function of the radial position is:

$$u_{\mathrm{a}}(t) = U_0 - \dot{d}(t) \cdot p_y, \tag{7}$$

where $p_y$ is the distance on the $y$-axis from the rotor apex.

## 4   Data processing

The tower-top loads were post-processed to obtain the aerodynamic rotor loads. These tower-top loads include forces from rotor aerodynamics and the rotor-nacelle assembly inertia and weight. Inertia and weight contributions were subtracted from the recorded loads using the post-processing method described by Fontanella et al. (2022a), isolating the aerodynamic forces. The aerodynamic loads are represented in the hub reference frame ($x - y - z$).

The aerodynamic rotor loads were analyzed both in the frequency domain and the time domain. The zero-peak amplitude of the aerodynamic loads was computed as the fast Fourier transform (FFT) amplitude at the frequency of platform motion. The phase shift between the aerodynamic rotor load and the platform motion was computed based on the real and imaginary part of the complex FFT at the frequency of interest.

In the time domain, rotor loads exhibit a response at the blade-passing frequency caused by rotor mass imbalance and at higher harmonics resulting from aerodynamic imbalance. The loads were low-pass filtered with a 3 Hz cut-off frequency to eliminate the effects of imbalance. This process was performed in the frequency domain by computing the complex FFT, keeping the frequency components up to 3 Hz, and then utilizing the inverse FFT to reconstruct the signal in the time domain. Filtering based on FFT and IFFT was selected over other techniques because it eliminates the harmonics at the rotor spinning frequency (4 Hz) due to imbalance, without affecting the response at 2 Hz, which is the highest frequency of platform motion considered in the experiment. The filtered loads include approximately 40 periods of platform motion. Subsequently, these loads were binned based on platform motion and phase averaged.

Velocity measurements using a hot wire were taken at a sampling frequency of 10 kHz. To prevent high-frequency electrical noise from skewing the turbulent statistics, the data were digitally filtered with a low-pass filter with a cut-off frequency of 100 Hz. The filter cut-off frequency was chosen carefully after analyzing the turbulent scales to identify the smallest and highest-frequency energy-containing scales, ensuring accurate estimation of the turbulent kinetic energy. By calculating the auto-correlation function of the stream-wise velocity at all measurement points, we observed that the smallest and highest-frequency scales appear in the undisturbed flows approaching the turbine and in the free-stream areas beside the rotor wake. In these low-turbulence zones, the autocorrelation consistently shows a correlation time of about 0.12 s, suggesting a correlation scale of around 0.5 m and an integral length scale of around 0.1 m. Filtering at 100 Hz captures the most energetic scales, along with a significant part of the inertial range of the energy spectrum visible in Fig. 2. Additionally, 100-Hz filtering in free-stream regions estimated a turbulence intensity of 1.5-2%, aligning with flow measurements conducted before testing.

Within the wake of the wind turbine, the energy-containing scales were observed to be larger in size and lower in frequency, with sufficient magnitude to dominate electrical noise. Thus, although the filter was not strictly necessary, it was applied consistently across all regions of the flow.

The acquisition time for each measurement point corresponded to 12 platform motion cycles. A sensitivity study conducted at the beginning of the test campaign determined that 12 cycles are adequate for achieving convergence of mean wind speed, turbulence intensity, and velocity spectrum amplitude at the frequency of platform motion. This is considered a reasonable compromise between the accuracy of parameter estimation relevant to this study and the time required to conduct the measurements. The velocity time series were binned according to platform motion and phase-averaged, in a similar manner to the loads.

## 5 Results

This section presents the findings obtained from a subset of the scenarios introduced in Sect. 3 that was found suitable to showcase the most interesting phenomena seen in the experiment. Section 5.1 reports the response of aerodynamic rotor loads to platform motion in different directions. Section 5.2 explores how platform surge and pitch motions influence the wake, Sect. 5.3 examines the impact of yaw motion, and Sect. 5.4 discusses the effect of crosswind motion from combined surge and sway.

### 5.1 Aerodynamic rotor loads

The rotor is perpendicular to the wind tunnel floor and the wind field is uniform in space. When the wind turbine tower base is fixed, the aerodynamic rotor loads are only the thrust force ($F_x$) and the torque ($M_x$), which were equal to $36.47\,\mathrm{N}$ and $2.97\,\mathrm{N\,m}$, respectively. To facilitate comparison with other studies, rotor loads are also presented using non-dimensional coefficients. The thrust coefficient is

$$C_t = \frac{F_x}{\frac{1}{2}\rho U_0{}^2 \pi R^2},$$
(8)

where $\rho$ is the air density and $R$ the rotor radius. The torque coefficient is given by Eq. 9

$$C_q = \frac{M_x}{\frac{1}{2}\rho U_0{}^2 \pi R^2 R}.$$
(9)

With a fixed tower base, the thrust coefficient was $C_t = 0.86$ and the torque coefficient was $C_q = 0.06$.

Rotor loads oscillate when the wind turbine experiences platform movement. For surge motion, the rotor axis stays aligned with the wind direction and $F_x$ and $M_x$ are the only aerodynamic loads. Pitch motion and movements resulting from the combination of surge and sway or pitch and roll, result in a skewed flow. In these cases, there are aerodynamic loads in different directions ($F_x, F_y, F_z, M_x, M_y, M_z$). However, the amplitude of the loads different from the thrust and torque (i.e., $F_y, F_z, M_y, M_z$) are generally small.

Figure 4 shows the rotor thrust and torque for pitch motion at three different amplitude values, with a frequency of 1 Hz. The loads were obtained from measurements using the data processing method described in Sect. 4, and the resulting phase-averaged time series are presented over one period of platform motion in the angle domain, using platform-motion phase instead of time. The smooth time series indicate that the number of motion cycles included in the measurements and the data processing method effectively isolate the aerodynamic loads response to platform motion. The average values of $M_x$ match the result with fixed-tower base. In cases with $a = 0.6°$ and $a = 1.9°$, the average $F_x$ is slightly lower than in the fixed case, but it matches the fixed-tower base case when $a = 1.3°$. Variations in the mean value of $F_x$ are likely attributable to slight differences in zero blade-pitch recalibration performed during testing. $F_x$ and $M_x$ display a sinusoidal pattern at the same motion frequency, with an approximate $90°$ phase shift relative to the platform displacement. This proves that rotor load fluctuations are mainly due to variations in apparent wind speed caused by platform movement as observed in previous experiments (Fontanella et al., 2021).

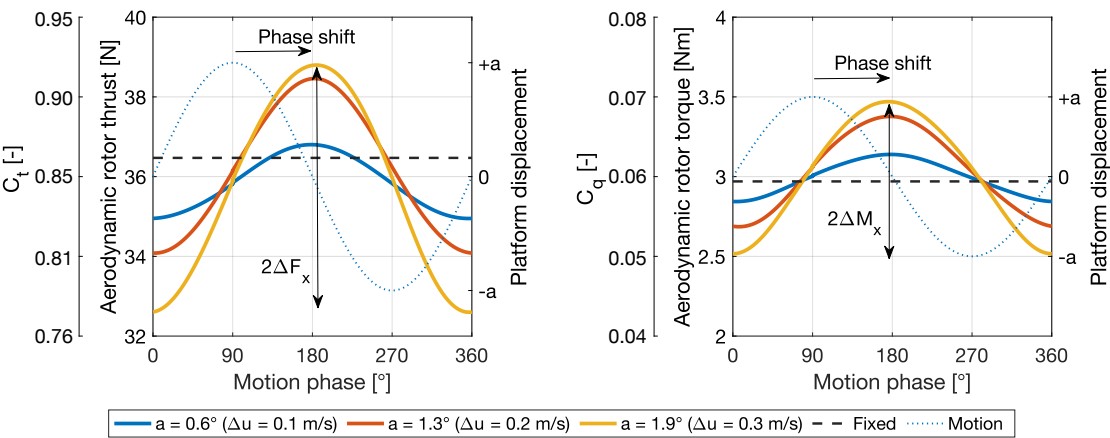

**Figure 4.** Phase-averaged time series of aerodynamic rotor thrust and torque for platform pitch motion at a frequency of 1 Hz, which corresponds to a reduced frequency of 0.6, with three different motion amplitudes ($a$). "Fixed" refers to the scenario with fixed tower base. $\Delta u$ is the amplitude of apparent wind changes caused by platform motion; $\Delta F_x$ and $\Delta M_x$ denote the amplitudes of the thrust and torque oscillations, respectively.

To compare loads obtained in the different scenarios involving motions in the surge, sway, roll, and pitch directions, the zero-peak amplitude of thrust and torque, $\Delta F_x$ and $\Delta M_x$ respectively, were normalized according to the hub motion amplitude in the $x_i$ direction ($d_{\mathrm{hub},x}$) in meters. Figure 5a illustrates the variation in normalized aerodynamic rotor thrust, while Figure 5b presents the variation in normalized aerodynamic rotor torque as the platform motion frequency changes. A linear regression is fitted to the measurements and is included in Figure 5a and Figure 5b. The loads with motion frequencies of 0.5 Hz and 1 Hz change linearly with respect to the platform motion amplitude, as evidenced by the normalized points aligning with the regression line; additionally, the loads exhibit a linear increase with frequency. Thus, at 0.5 Hz and 1 Hz, the rotor aerodynamic loads change linearly with apparent wind variations in the $x_i$ direction ($\Delta u$). For example, increasing the amplitude of platform

in the $x_i$ direction or the frequency by a factor of 2 would result in changes in rotor aerodynamic loads of the same order. The variation in loads for motion cases at a frequency of 2 Hz exceeds the expected linear trend. For instance, when the surge motion has an amplitude of 0.048 m and a frequency of 2 Hz, corresponding to the maximum $\Delta u$ investigated in the experiment, the amplitude of thrust force oscillation is 6.99 N and not the 5.6 N predicted by linear regression. The deviation of aerodynamic loads from the linear trend at 2 Hz was observed also by Bergua et al. (2023), but it was not reflected in the high-fidelity

simulations of the experiment conducted alongside the measurements analysis. Numerical simulations of the wind turbine scale model utilized in this experiment indicate that the amplitude of aerodynamic load oscillations decreases below the linear trend as the frequency increases beyond 2 Hz (Ribeiro et al., 2023). Thus, the higher-than-linear response at a motion frequency of 2 Hz is expected to be due to non-idealities in the experimental setup, primarily fluctuations in rotor speed, with an amplitude up to 0.6 rpm, and a minor flexible response of the tower, where the amplitude of nacelle motion is up to 7% higher than that of the

platform base motion. These non-idealities should be taken into account when comparing the experimental load response with the results of numerical simulations. However, they are considered small enough to not impact the observed wake dynamics trends.

       Figure 5c and Figure 5d show the phase shift of the aerodynamic rotor loads with respect to the platform motion. The phase shift is approximately $90°$, indicating that the loads oscillations are in phase with the fluctuations in apparent wind speed.

The loads response is consistent with the behavior observed in previous studies investigating platform surge and pitch motions (Bergua et al., 2023).

       Yaw motion leads to oscillations in the aerodynamic yaw moment $M_{z,p}$ about the $z_p$ axis. The yaw moment coefficient $C_{mz}$ is calculated in the same way as the torque coefficient, with $M_x$ replaced by $M_{z,p}$ in Eq. 9. Figure 6 shows the aerodynamic rotor thrust and the aerodynamic yaw moment in one cycle of yaw motion for four yaw motion scenarios. The yaw moment

exhibits an average value slightly different from zero, probably because of a small misalignment of the wind turbine during the tests. The loads are affected by oscillations at the frequency of platform motion. These oscillations are determined by the apparent wind speed created by platform movement $u_a$, rather than by the misalignment of the rotor relative to the incoming wind, since the yaw angle is relatively small. Rotor thrust oscillations exhibit much smaller amplitudes compared to those observed in surge, sway, roll, and pitch motions. In the 0.5 Hz scenario, the oscillations are minimal and almost indistinguishable

from measurement noise, resulting in the irregular pattern shown in the figure. Across all four scenarios, the average thrust is marginally lower than that of the fixed turbine case.

## 5.2   Wake with platform pitch and surge motions

The analysis of aerodynamic rotor loads indicates that pitch and surge movement leads to periodic thrust changes driven by apparent wind speed fluctuations caused by the motion. These thrust changes perturb the wake of the wind turbine. The effect

is assessed by comparing the average velocity and turbulence intensity of the floating case with the bottom-fixed case, and examining velocity fluctuations coherent with platform motion.

       Figure 7a shows the average velocity profile of the wake at hub height and $x_w = 3D$, for the wind turbine with a fixed foundation and under four platform pitch motion conditions. The velocity profile has a double-Gaussian shape. In line with

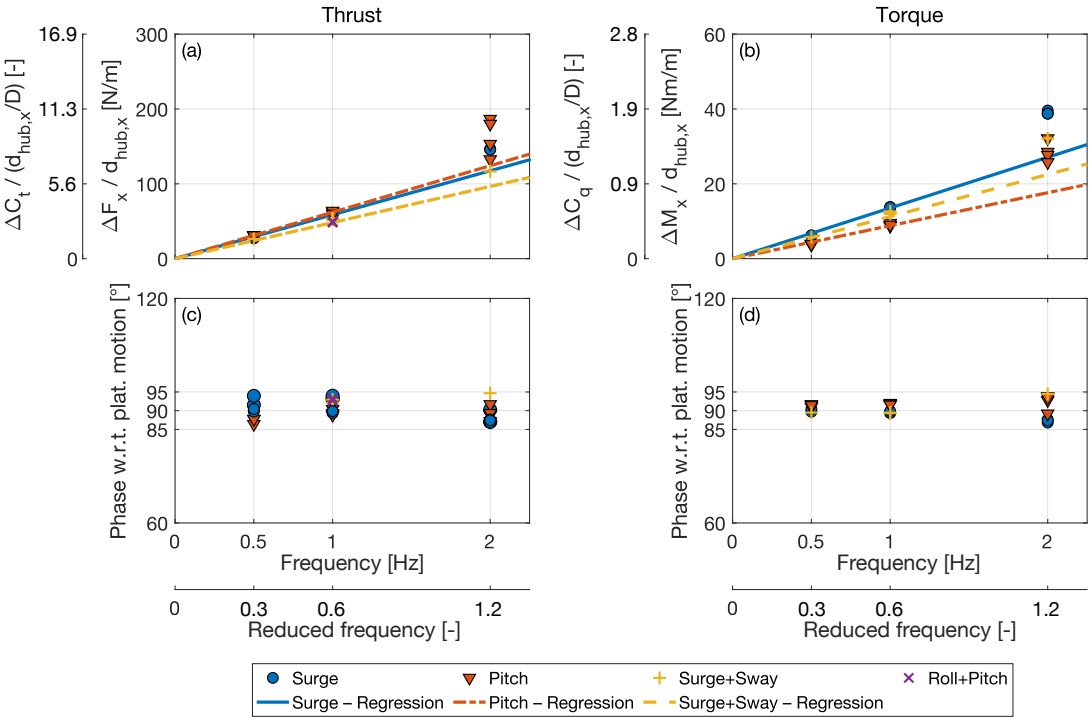

**Figure 5.** Normalized variation of the aerodynamic rotor thrust **(a)** and phase shift with respect to platform motion **(c)**. Normalized variation of the aerodynamic rotor torque **(b)** and phase shift with respect to platform motion **(d)**.

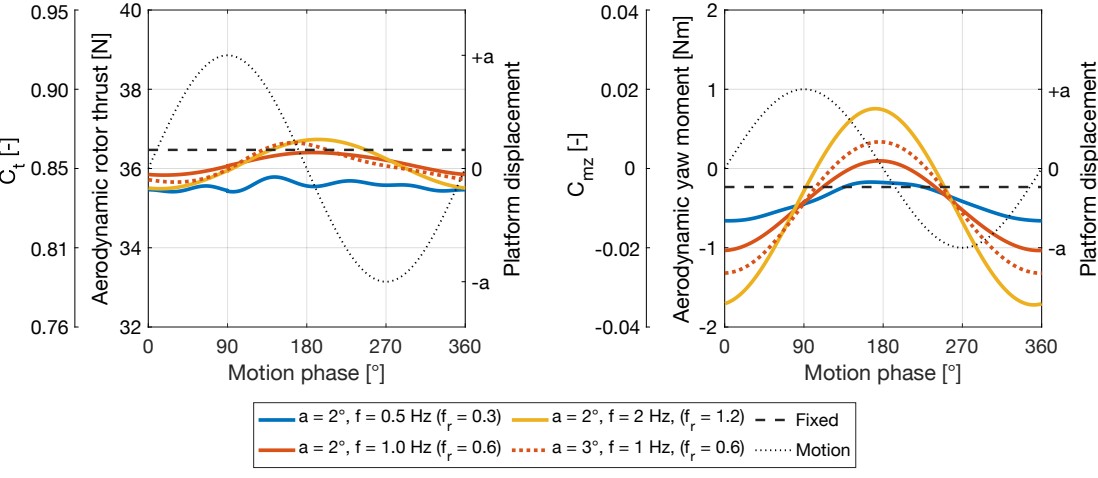

**Figure 6.** Phase-averaged aerodynamic rotor thrust and yaw moment (about the $z_\mathrm{P}$ axis) time series during yaw motion.

previous measurements of the wake of the same wind turbine of Fontanella et al. (2022b), the velocity profile is asymmetric relative to the rotor axis. The wake center is shifted approximately $0.06D$ toward the negative $y_\mathrm{i}$-axis, and the velocity minima peaks on each side of the wake differ in shape. In the experimental study conducted by Pierella and Sætran (2017), it was observed that the interaction between the turbine tower wake and rotor-induced flow results in wake asymmetry. The tower creates a low-velocity zone that is entrained into the wake and rotates with it, persisting up to $3D$ downstream. The mean velocity profile remains largely unchanged by platform motion, with only minor variations observed in the peak of minimum velocity on the positive side of the $y_\mathrm{i}$-axis.

The turbulence intensity in the wake is computed as:

$$I(y_\mathrm{i}) = \frac{\sigma_u(y_\mathrm{i})}{U(y_\mathrm{i})}, \tag{10}$$

where $\sigma_u(y_\mathrm{i})$ represents the standard deviation of the local streamwise velocity, while $U(y_\mathrm{i})$ denotes its time-average value. Since $U(y_\mathrm{i})$ is nearly the same in both the platform motion and fixed tower base scenarios, any increase in $I$ is due to greater variability of the streamwise velocity. The turbulence intensity profiles are reported in Figure 7b. They all show two peaks in correspondence of the rotor edges that are attributed to the wake shear layer. At a platform pitch motion frequency of 1 Hz, the turbulence intensity is increased near the center of the wake, indicating that the mixing layer shifts towards the wake core. This behavior intensifies as the amplitude of motion at a specific frequency increases, causing both apparent wind speed and thrust oscillations to rise.

Figure 7c shows the streamwise velocity at $x_\mathrm{w} = 3D$ and $y_\mathrm{i} = 0.6\,\mathrm{m}$ obtained from phase-averaging of velocity time series acquired over 12 platform motion cycles, as described in Sect. 4. The point at $y_\mathrm{i} = 0.6\,\mathrm{m}$ is selected because it represents half the rotor radius, where the blade undergoes the greatest local normal force variation (Fontanella et al., 2022b), and it does not align with the wake shear layer in the fixed-turbine scenario. Significant velocity fluctuations occur at this location when the wind turbine undergoes platform movement, thus $y_\mathrm{i} = 0.25D$ is used for comparing different motion conditions. The data points obtained from phase-averaging were fitted with a sine function with a frequency equal to the motion frequency to emphasize the velocity oscillations that match the platform motion. The determination coefficient $R^2$ is used to indicate how well the sinusoidal function fits the data points. In the cases examined in Fig. 7c, the fitted sine function captures a significant portion of the velocity variability, indicating the presence of flow structures in the wake that are strongly coherent with the sinusoidal platform motion. Similar dynamics have been reported by Wei et al. (2024), who demonstrated that periodic upstream forcing, such as thrust variations caused by platform motion, generates traveling-wave undulations in the wake radius and streamwise velocity.

For the same motion amplitude, the largest velocity oscillations are with $f = 1$ Hz despite this condition having lower thrust variation amplitude (see Fig. 5) than at $f = 2$ Hz. When the motion frequency is 2 Hz, oscillations are observed only at the boundaries of the wake (approximately at $y_\mathrm{i} \pm 0.45D$) that corresponds to the location of the shear layer. Coherent flow structures are mostly absent at other locations. As shown in the graph, the sine wave at the motion frequency does not fit the velocity data as well as it does in the 0.5 Hz and 1 Hz cases. Research on dynamic induction control has shown that the evolution of the wake is influenced by the reduced frequency of periodic fluctuations in thrust force (Munters and Meyers,

2018). We assume that similar effects occur in the wake of a floating wind turbine where fluctuations in thrust force are due to the movement of the platform. Among the conditions explored in this study, the strongest perturbation of the wake occurs when the reduced frequency of platform motion is 0.6. This agrees with the findings of Messmer et al. (2024a) who observed in a wind tunnel experiment that optimal wake recovery occurs with surge motions at a reduced frequency range of 0.5 to 0.8.

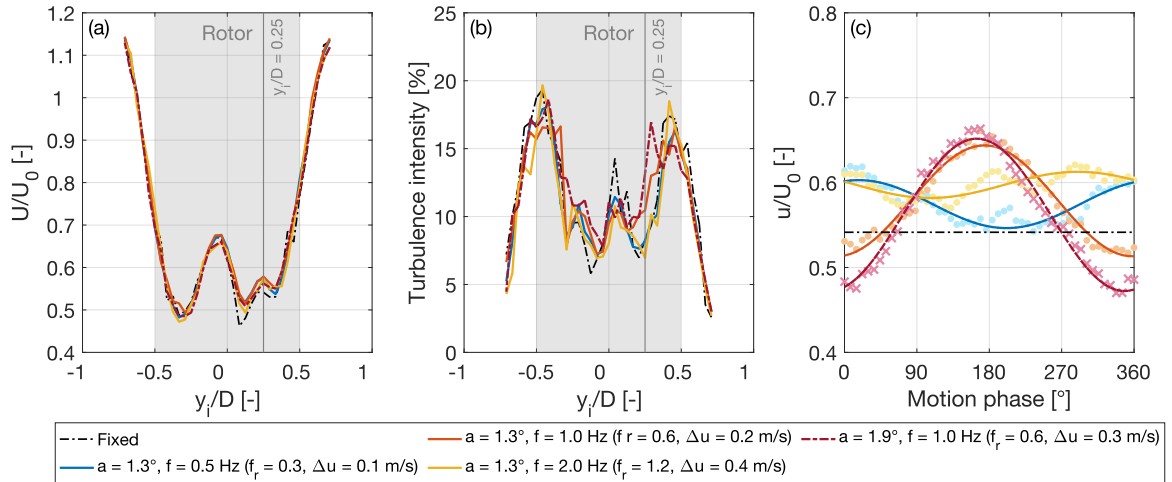

**Figure 7.** Wake at hub height and $x_\mathrm{w} = 3D$ with platform pitch motion. **(a)** Mean streamwise velocity normalized by the free-stream wind speed ($U_0$). **(b)** Turbulence intensity. **(c)** Phase-averaged time series of streamwise velocity $u$ at $y_\mathrm{i} = 0.25D$: the solid lines highlight the first-order sinusoid at the motion frequency that was fitted to the data points obtained from phase-averaging and $R^2$ is the coefficient of determination assessing the fit quality.

Velocity fluctuations coherent with platform motion vary throughout the wake width. This is examined in Fig. 8 that shows the phase-averaged velocity in with pitch motion at $x_\mathrm{w} = 3D$ and at various points along the $y_\mathrm{i}$-axis. With motion frequencies of 0.5 Hz and 1 Hz, coherent fluctuations are not observed at the rotor edges, where the shear layer is located, and in the wake center. At $y_\mathrm{i} = \pm 0.375D$, the velocity oscillations have similar amplitude and phase shift relative to the platform motion. In other positions within the inner region of the wake, the amplitude of velocity oscillations and their phase in relation to platform motion are not symmetrical with respect to the rotor axis. The development of coherent flow structures is closely associated with their interaction with other turbulent structures and the turbulent mixing process. Variations in the background flow, as shown in Fig. 2, and interaction of the rotor wake with the tower wake may explain the observed asymmetry of velocity fluctuations relative to the rotor axis.

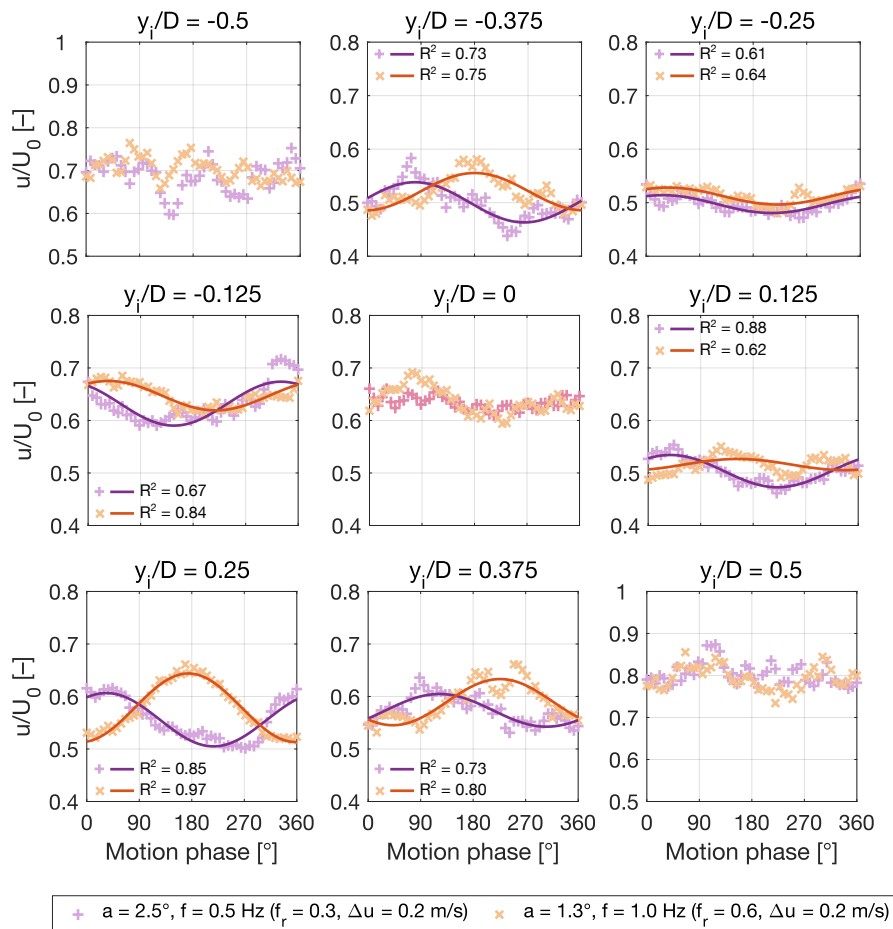

**Figure 8.** Phase-averaged time series of streamwise velocity $u$ normalized by free-stream velocity $U_0$ during pitch motion with $\Delta u = 0.2\,\mathrm{m\,s^{-1}}$, $f = 0.5$ Hz and $f = 1$ Hz at $x_i = 3D$. The solid lines highlight the first-order sinusoid at the motion frequency that was fitted to the data points obtained from phase-averaging and $R^2$ is the coefficient of determination assessing the fit quality. A satisfactory fit of the sinusoid was not achieved in the positions $y_i = \pm 0.5D$ and $y_i = 0$.

The analysis carried out in Fig. 7 for the wake at $x_w = 3D$ are repeated in Fig. 9 for the wake at $x_w = 5D$. The average velocity exhibits a Gaussian profile that is mostly unchanged by platform motion. The wake velocity in the rotor region (for $|y_i| \leq 0.5D$) is known at $N_p$ points in the $y_i$ direction that are symmetrically distributed with respect to the $x_i$-axis; the average velocity in this one-dimensional discrete domain is computed as:

$$U_{\mathrm{avg}} = \frac{\sum_{j=1}^{N_\mathrm{P}} |r_j| U_j}{\sum_{j=1}^{N_\mathrm{P}} |r_j|}, \tag{11}$$

where $U_j$ is the time-average velocity at the j-th point which is located at a distance $r_j$ from the $x_i$-axis. In the rotor region, $U_{\mathrm{avg}}$ with motion is approximately 65% of $U_0$, slightly exceeding the average of 62.5% of $U_0$ in the fixed condition.

The turbulence intensity exhibits the two peaks associated with the shear layer, that are situated closer to the wake center than at $x_w = 3D$. In line with findings at $x_w = 3D$, the pitch motion scenario at $f = 1$ Hz exhibits higher turbulence at the wake center compared to other scenarios. The shear layer occupies a larger part of the wake compared to the fixed case, thus the transition from near to far wake appears to be accelerated and takes place closer to the rotor.

As shown in Fig. 9c, the velocity oscillations that are coherent with platform motion are less pronounced at $x_w = 5D$ than at $x_w = 3D$. This is evidenced by $R^2$, which indicates that the fitted sine functions capture less of the variability of the velocity.

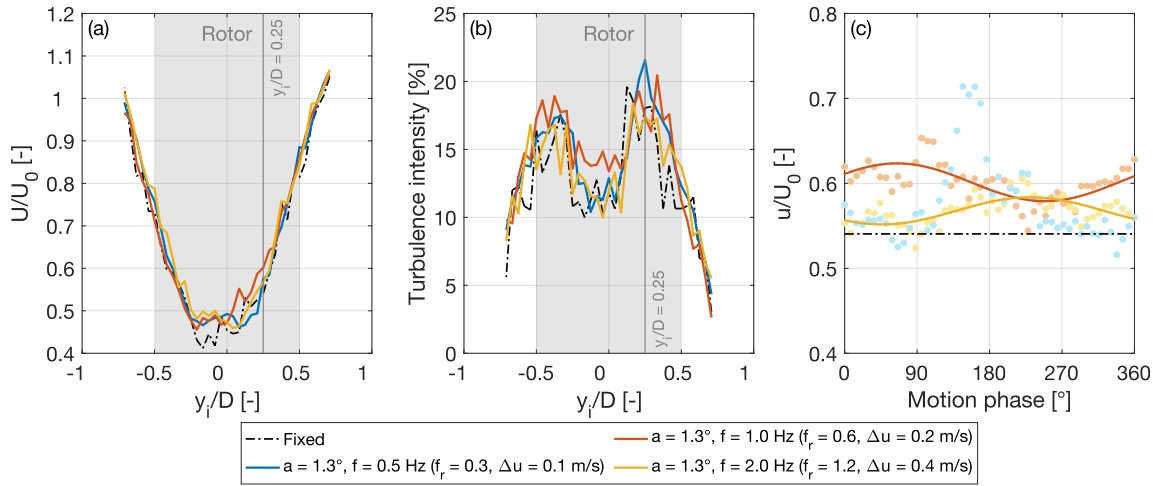

**Figure 9.** Wake at hub height and $x_w = 5D$ with platform pitch motion. **(a)** Mean streamwise velocity normalized by the free-stream wind speed ($U_0$). **(b)** Turbulence intensity. **(c)** Phase-averaged time series of streamwise velocity $u$ at $y_i = 0.25D$: the solid lines highlight the first-order sinusoid at the motion frequency that was fitted to the data points obtained from phase-averaging and $R^2$ is the coefficient of determination assessing the fit quality. A satisfactory fit of the sinusoid was not achieved with $f = 2$ Hz.

The progression of the turbulence components in the wake with increasing distance from the rotor is analyzed in Fig. 10, which presents the power spectral density (PSD) and the phase-averaged streamwise velocity for the scenario involving a pitch motion with an amplitude of $1.3°$ and a frequency of $1$ Hz. At $x_w = 3D$ the PSD displays peaks at the platform motion frequency corresponding to velocity oscillations observed in the phase-averaged time series. Flow structures coherent with the periodic platform movement dominate the wake response and a distinct increase and decrease in velocity, matching the periodicity of platform motion, is observable on both sides of the wake. At $x_w = 5D$, the peak in the PSD at the frequency of motion becomes less evident. As the wake begins to dissipate and turbulence develops, there is an increase in spectral energy from $x_w = 3D$ to $x_w = 5D$ across the wake width and over a broad frequency range. As energy builds up across different frequencies, the direct influence of platform motion on the wake becomes partially obscured by other turbulent structures. This is evidenced by the lower coherency of velocity oscillations with the platform motion at $y_i = 0.25D$, as shown by the sine function fit in Fig. 9, and the reduced smoothness of the phase-averaged wake in Fig. 10d.

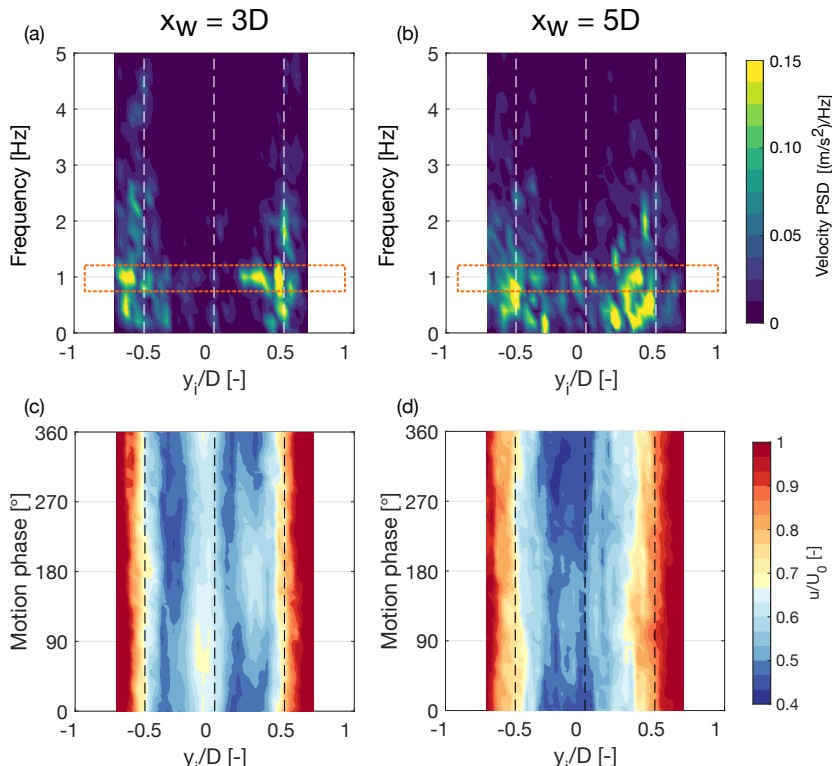

**Figure 10.** Progression of the turbulence components in the wind turbine wake at increasing distances ($x_{\mathrm{w}}$) from the rotor during pitch motion with $a = 1.3°$ and $f = 1$ Hz. **(a, b)**: power spectral density (PSD) of streamwise velocity. **(c, d)**: phase-averaged streamwise velocity. Vertical dashed lines indicate the rotor axis and edges.

Figure 11 compares, at $x_{\mathrm{w}} = 3D$, the wake with surge and pitch motions. Both motion conditions result mainly in thrust oscillations at the frequency of platform motion and are characterized by the same reduced frequency and $\Delta u$. Their influence on the wake is comparable in terms of average velocity (at $x_{\mathrm{w}} = 5D$, $U_{\mathrm{avg}}$ is 65% of $U_0$ in both cases) and turbulence intensity. Motion in the pitch direction causes larger velocity fluctuations, attributed to the small skew of the inflow over the rotor that is absent in surge motion.

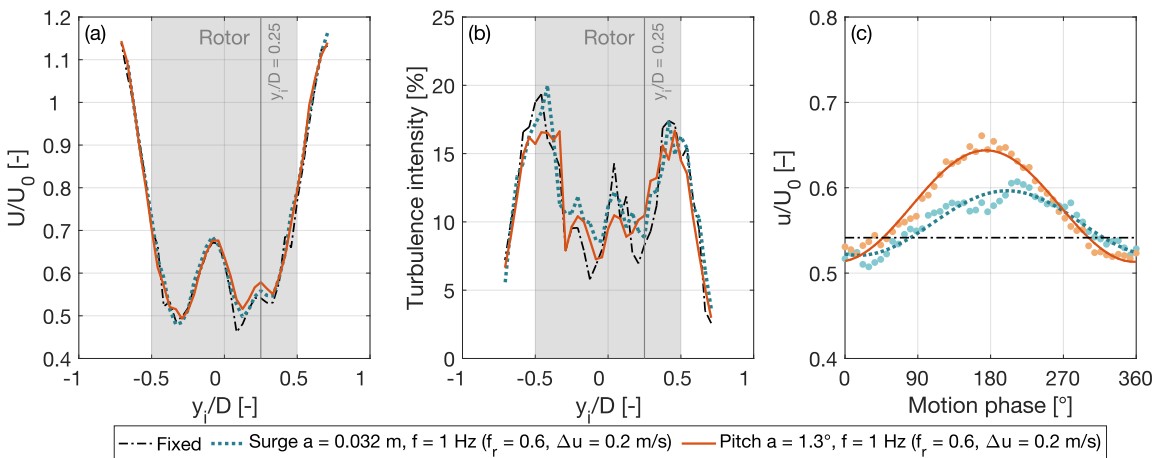

**Figure 11.** Wake at hub height and $x_w = 3D$ with platform surge and pitch motions of frequency 1 Hz. The surge and pitch motions are associated with the same amplitude of apparent wind speed $\Delta u = 0.2\,\mathrm{m\,s^{-1}}$. **(a)** Mean streamwise velocity normalized by the free-stream wind speed ($U_0$). **(b)** Turbulence intensity. **(c)** Phase-averaged time series of streamwise velocity at $y_i = 0.25D$: the solid lines highlight the first-order sinusoid at the motion frequency that was fitted to the data points obtained from phase-averaging and $R^2$ is the coefficient of determination assessing the fit quality.

Figure 12a-b shows the wake along the $z_i$-axis at $x_w = 3D$ for both surge and pitch motions at $f = 1$ Hz with the same $\Delta u$. The velocity over one motion cycle is almost identical, and in pitch the wake center moves only slightly in the vertical direction. Figure 12c compares the average velocity along the $z_i$-axis at $x_w = 3D$ with different pitch motions. Like the horizontal profile of Figure 7a, the vertical velocity profile exhibits a double-Gaussian shape and is largely unaffected by platform movements.

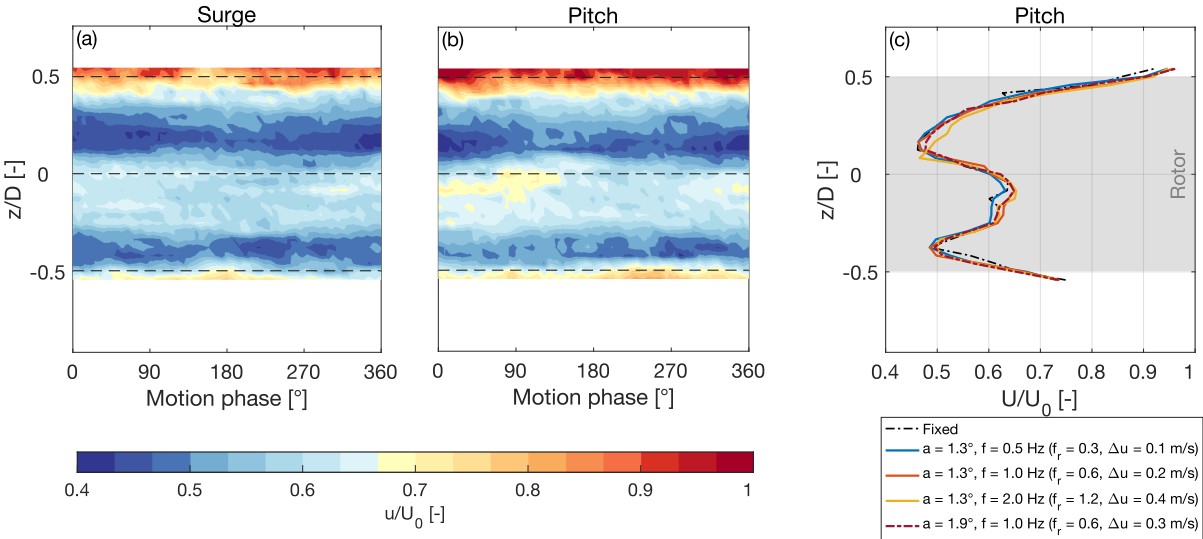

**Figure 12.** Wake along the $z_i$-axis and $x_w = 3D$ with platform surge and pitch motions. **(a)** Phase-averaged streamwise velocity during surge motion with $a = 0.032\,\mathrm{m}$, $f = 1\,\mathrm{Hz}$ (the dashed lines are in correspondence of the rotor axis and the rotor edges). **(b)** Phase-averaged streamwise velocity during pitch motion with $a = 1.3°$, $f = 1\,\mathrm{Hz}$ (the dashed lines are in correspondence of the rotor axis and the rotor edges). **(c)** Mean streamwise velocity normalized by the free-stream wind speed ($U_0$) along the $z_i$ axis with different pitch motions.

### 5.3 Wake with platform yaw motion

Movement in the yaw direction generates different aerodynamic loads compared to those created by surge or pitch motions. While the thrust variations are minimal, there is an oscillating yaw moment on the rotor. Figure 13 examines the development of the wake at $x_w = 3D$ with platform yaw motion. The average velocity is unchanged compared to the fixed scenario. Turbulence intensity increases near the wake center and decreases close to the rotor edges, showing that the shear layer moves toward the wake center, a pattern also noted with surge and pitch motions. However, this is not the effect of periodic fluctuations in thrust

force that are almost absent with movement in the yaw direction (see Fig. 6).

Figure 13c shows the velocity over one cycle of yaw motion with $a = 2°$, $f = 1\,\mathrm{Hz}$. The wake undergoes lateral meandering which is noticeable when tracking the relative maximum velocity at the center and the absolute minimum velocity on the left side of the double-Gaussian (black lines in Fig. 13c). The wake lateral movement matches the platform frequency and shows strong correlation across the wake width. We attribute wake meandering to the periodic yaw moment on the rotor. The

meandering reaches its peak amplitude when $f = 1\,\mathrm{Hz}$, is reduced when $f = 0.5\,\mathrm{Hz}$, and nearly disappears when $f = 2\,\mathrm{Hz}$. Thus, this wake response is also sensitive to the frequency of the oscillations of the rotor loads and has the highest intensity when $f_r$ is equal to 0.6. At $x_w = 5D$, the meandering disappears. With a yaw motion of $a = 2°$, $f = 1\,\mathrm{Hz}$, $U_{avg}$ is equal to 65.8% of $U_0$ at $x_w = 5D$, showing a 5% increase over the fixed-bottom case.

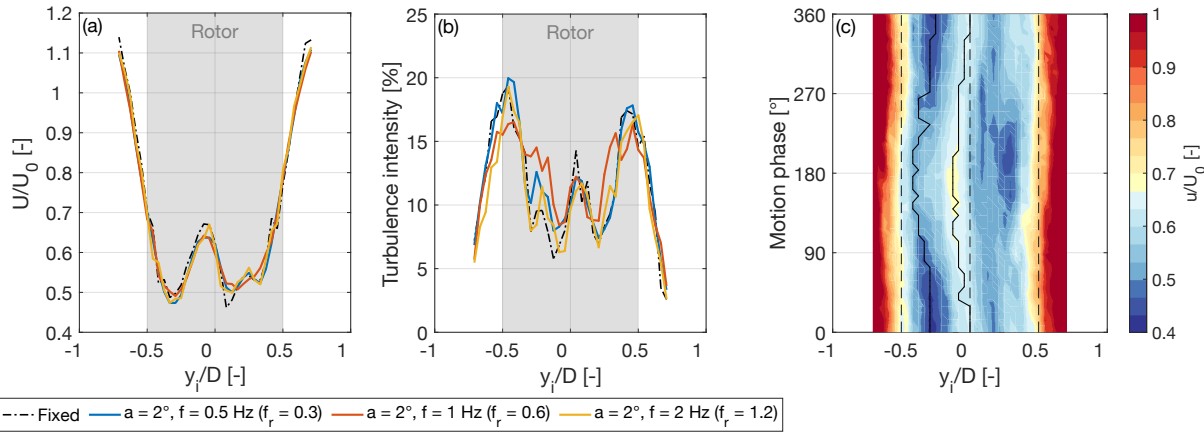

**Figure 13.** Evolution of the wake at hub height and $x_{\mathrm{w}} = 3D$ with platform yaw motion. **(a)** Mean streamwise velocity normalized by the free-stream wind speed ($U_0$). **(b)** Turbulence intensity. **(c)** Phase-averaged streamwise velocity during yaw motion with $a = 2°$, $f = 1$ Hz (the dashed lines are in correspondence of the rotor axis and the rotor edges).

## 5.4 Wake with platform crosswind motion

Whenever movement in the crosswind direction takes place, thus combining both surge and sway, the apparent wind is skewed as depicted in Fig. 3. The thrust exhibits periodic oscillations that are proportional to the rotor forward motion in the $x_{\mathrm{i}}$ direction, similar to those observed in cases of pure surge motion. Additionally, throughout a motion cycle, the wake is released from various positions along the $y_{\mathrm{i}}$-axis.

Figure 14 shows the wake at $x_{\mathrm{w}} = 3D$ with motions that are the combination of surge and sway. The platform translation 440 of $a = 0.032$ m and $f = 1$ Hz is at an angle $\gamma$ ranging from $0°$ (i.e., pure surge motion) to $45°$. When $\gamma = 45°$, the near wake has lateral meandering that is instead absent in the case with $\gamma = 0°$. The onset of wake meandering is attribute to the lateral component of the apparent wind $v_{\mathrm{a}}$ and to the periodic release of the wake at different $y_{\mathrm{i}}$ positions, which do not occur in the surge motion case. The wake lateral movement matches the platform frequency and is correlated across the wake width. During one period of platform motion, the minimum velocity on the left side undergoes a lateral displacement of $0.13D$, which 445 is larger than the rotor crosswind motion that has a peak-to-peak amplitude of $0.019D$ (corresponding to $0.045$ m). The large-eddy simulation study of a wind turbine wake under side-to-side motion of Li et al. (2022) shows that this movement can cause an instability in the wake shear layer, which can propagate and amplify downstream the rotor. This can result in wake lateral meandering amplitudes that are up to an order of magnitude larger than the rotor lateral motion, as observed in this experiment.

With surge and sway motion, the turbulence intensity in the wake center increases more significantly than with surge mo-450 tion alone. This increase becomes more pronounced as the angle $\gamma$ grows, as shown in Fig. 14c. The combination of wake meandering and velocity fluctuations appears to further promote the transition from near to far wake.

At $x_{\mathrm{w}} = 5D$, the meandering and the velocity fluctuations disappear. With a motion of $a = 0.032$ m and $f = 1$ Hz, with $\gamma = 30°$, the average wake velocity at $x_{\mathrm{w}} = 5D$ is 63.5% of $U_0$.

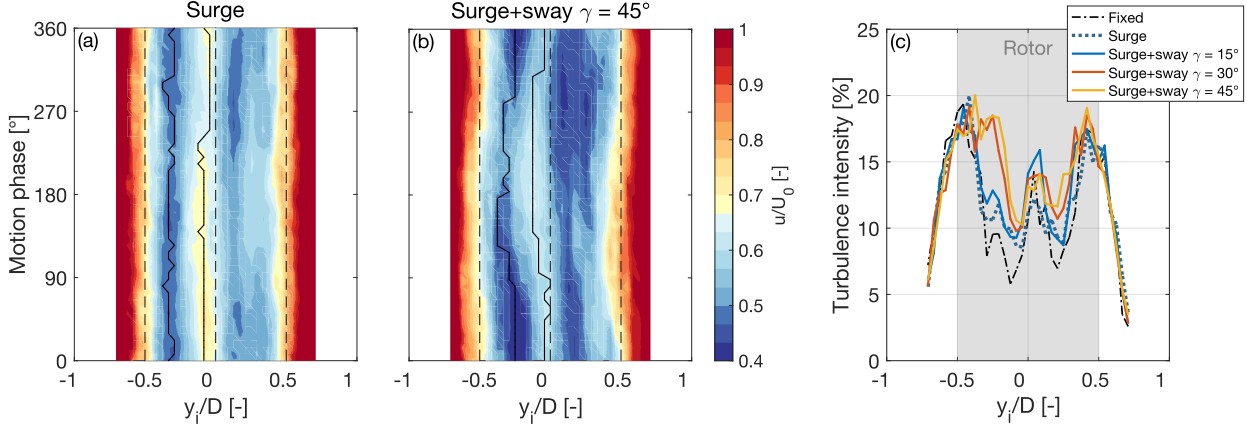

**Figure 14.** Wake at hub height and $x_i = 3D$ with a platform motion that is the combination of surge and sway. **(a)** Phase-averaged streamwise velocity in pure surge motion of $a = 0.032$ m and $f = 1$ Hz. **(b)** Phase-averaged streamwise velocity during a translation motion at angle $\gamma = 45°$ of $a = 0.032$ m and $f = 1$ Hz. **(c)** Turbulence intensity for translation motions with four values of $\gamma$ (the surge motion corresponds to $\gamma = 0°$).

## 6 Conclusions

This study investigated how platform motion in different directions affects rotor aerodynamics and correlated the rotor response to wake development. Wake measurements, taken with hot-wire probes from $3D$ to $5D$ downstream, were compared between scenarios with prescribed platform motions and the bottom-fixed case.

Depending on the direction of platform motion, we distinguished three conditions. With surge and pitch, the rotor moves in the wind direction; this makes both rotor thrust and torque fluctuate at the same frequency of motion. The oscillations in rotor
loads are driven by the apparent wind created by platform movement. The average values of thrust and torque remain relatively unaffected by the platform motion, indicating that the overall performance of a floating wind turbine is comparable to that of a fixed-bottom turbine. The thrust oscillations translate into fluctuations of the near wake velocity that peak with a motion reduced frequency of 0.6. Yaw motion leads to oscillations of the yaw moment at the rotor and to lateral wake meandering correlated with the platform motion frequency. Combining surge and sway motions results in skewed apparent wind, causing
lateral wake meandering that adds to the velocity fluctuations caused by oscillations in thrust force. In a floating wind farm, the velocity fluctuations in the wake can lead to fluctuating loads on downstream turbines. Upcoming wind tunnel tests with multiple scale models of floating turbines will investigate this further.

For the wind turbine operating conditions, motion scenarios, and distances from the rotor examined in this experiment, the mean velocity in the wake is comparable to that of the bottom-fixed case. Instead, turbulence intensity increases near the wake
center, suggesting a shift of the mixing layer towards the wake core. Increased turbulence is linked to higher wake mixing, which could mean a faster recovery beyond $5D$, the furthest distance examined in the experiment. Future studies should

examine the wake further downstream to verify whether the increased wake turbulence induced by motion results in quicker wake recovery.

Greater turbulence intensity in undisturbed wind enhances wake mixing and facilitates the transition to the far wake, helping to recover the free-stream speed. In offshore environments, turbulence intensity typically ranges from 3% to 6%, with sheared wind. This experiment used a 1.5% turbulence intensity and a nearly constant vertical velocity profile, thus future testing should examine the effects of more realistic turbulence and wind shear on the evolution of floating wind turbine wakes. The flow within a wind farm is further complicated by the interactions between the wakes of various turbines, each subjected to dynamic platform motion. This condition is recommended for exploration in future experimental campaigns.

All experimental results obtained in this research aim to provide a reliable benchmark to validate numerical tools and to improve their modeling capabilities. Knowledge gained from experiments, along with better simulation tools can be leveraged to optimize future large-scale floating wind farms.

*Data availability.* Measurement data of the wind tunnel experiment are accessible at https://doi.org/10.5281/zenodo.13994980.

## Appendix A: Assessment of the wind turbine scale model aerodynamic design

The aerodynamic design of the wind turbine scale model was verified through a dedicated wind tunnel campaign, described in detail by Bayati et al. (2017b), that measured rotor thrust and torque at wind speeds up to $3.8\,\mathrm{m\,s^{-1}}$ (model scale) within the below-rated region. In this region, the rotor speed was adjusted to achieve the design tip-speed ratio of 7.5, with a fixed blade pitch of $0°$. This condition corresponded to the one evaluated during the rotor design phase (Bayati et al., 2017b). The rotor aerodynamic performance was evaluated in a steady wind field with a turbulence intensity of 1.5%, which matches the inflow

conditions considered in this study.

    Figure A1 presents the rotor thrust and torque measured during characterization as well as those obtained in the fixed tower-base scenario of the current wind tunnel experiment. These results are compared to the DTU 10-MW values at model scale for different wind speeds. The rotor thrust and torque of the DTU 10-MW were obtained from the report of Bak et al. (2013), where they were calculated using HAWC2Stab simulations and a blade-element momentum model to compute the rotor aerodynamic

loads. The full-scale values were converted to model scale through dimensional analysis, utilizing a geometric scale factor of 1:75 and a velocity scale factor of 1:3. This results in a force scale factor of 1:50625 and a moment scale factor of 1:3796875.

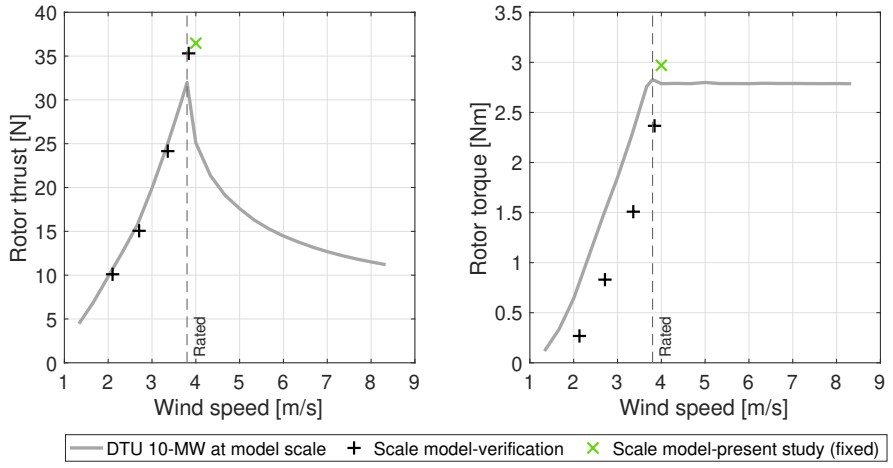

**Figure A1.** Steady-state rotor thrust and torque of the wind turbine model compared to the DTU 10 MW values at model scale for different wind speeds.

    The primary objective in designing the scale model rotor was to match the scaled thrust force of the reference wind turbine. This goal was successfully achieved, as the thrust produced by the scale model closely aligns with that of the reference turbine. However, the torque of the scale model is consistently lower than the scaled torque of the DTU 10-MW for below-rated

wind speeds. This discrepancy is due to the different aerodynamic characteristics—specifically, lift and drag—of the SD7032 low-Reynolds airfoil used in the scale model, compared to the airfoils employed in the DTU 10-MW blade. While the blade geometry was optimized to replicate the thrust behavior, simultaneously matching the torque was not feasible due to these inherent differences in airfoil performance.

At wind speeds close to the rated value, some discrepancies arise between the loads experienced by the full-scale wind turbine and those of the scale model. In this operating range, the full-scale turbine uses active blade pitching to maintain the rotor torque at its rated value, which also reduces rotor thrust. In contrast, the scale model was tested with a fixed blade pitch for simplicity, leading to the observed differences.

*Author contributions.* All authors prepared and conducted the experiment, and analyzed the measurement data. AF wrote the first draft of the article, while all authors contributed to its review and editing. MB, AB, and VD have procured the funding. MB, SM, AB, and VD have supervised the work.

*Competing interests.* At least one of the (co-)authors is a member of the editorial board of Wind Energy Science.

*Acknowledgements.* This research has been funded by the European Union – NextGenerationEU, M4C2 I1.1, Progetto PRIN 2022 "NET-TUNO", Prot. 2022PFLPHS, CUP D53D23003930006.

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
