# Peer review of "Wake Development in Floating Wind Turbines: New Insights and Open Dataset from Wind Tunnel Experiments"

_Wind Energy Science, 2024_

## Referee Comment (RC1)

**REVIEW OF WES-2024-140**

*Wake Development in Floating Wind Turbines: New Insights and Open Dataset from Wind Tunnel Experiments.*

*authors:*
Alessandro Fontanella, Alberto Fusetti, Stefano Cioni, Francesco Papi, Sara Muggiasca, Giacomo Persico, Vincenzo Dossena, Alessandro Bianchini, and Marco Belloli

**Summary:**

The manuscript entitled "Wake Development in Floating Wind Turbines: New Insights and Open Dataset from Wind Tunnel Experiments" introduces some early results from a battery of wind tunnel experiments involving a modeled floating offshore wind turbine. The experiments show a great deal of care and highlight some of the aerodynamic changes in the wake that arise due to imposed, periodic platform motions. The complete suite of experiments produced more results than could be succinctly shared in a manuscript, so only a subset of the results are discussed within. While these preliminary results certainly indicate the quality and value of the data set, some of the analysis requires more attention before it can be considered publication quality. One of the main considerations overlooked in the current work is potential interaction between periodic turbulence statistics in the wake, and the imposed platform motion. These interactions could lead to harmonics visible in velocity measurements, which are currently shown in terms of the phase of the imposed platform motion. I also point out a few concerns about the signal processing approach used by the authors. See specific comments below.

**Comments:**

- "NETTUNO" — Acronym used before it is defined
- "pre-commercial stage" — Is this still true? There are commercial deployments of floating offshore turbines already.
- "Very recently, Özinan et al. (2024) studied the near wake of a 2 MW floating wind turbine and found no evident impact of wave-induced motions on the average velocity of the wake, partially contrasting theoretical speculations." — Does this undercut the motivation for the work? Please clarify how findings in this reference influence the NETTUNO project.
- "10 MW" and "15 MW turbine" — Please specify the model of turbine used as a basis of the scaled experiments.
- "The design of the blades and tower focused on maximizing stiffness to minimize their aeroelastic response." — Since measurement of structural loads are among the project goals, can the authors provide some more detail on how aeroelastic scaling was approached for the wind turbine model? The authors state that the goal is to maximize blade stiffness, but this is not likely to produce results that scale up to the 10 MW turbine model. More justification for this choice would help readers understand the approach.

- "ensuring the rotor was vertical to the wind tunnel floor" — Does this mirror the 10 MW reference turbine design? Typically, the tower is nominally vertical but the rotor is not. Perhaps it would be instructive to state here that aligning the shaft with the mean flow is an intentional simplification, to limit loads to the axial thrust and moment in the baseline case.
- "six-component loads" — Please be more specific about the loads measured in the introduction. This statement seems to indicate that only aggregate rotor forces and moments were measured. Were the tower bending or twist measured? Blade loads?
- "The wind velocity in the turbine wake was measured using a hot wire anemometer." — While a hotwire is certainly a good system for pointwise turbulence measurements, using a single probe in isolation makes coherence and spatial correlations impossible. With such a capable traverse system, why not make multiple measurements simultaeneously?
- "The resulting average extended uncertainties, at a 95% confidence level, are 0.17 m/s for flow speed and 0.4% for turbulence intensity." — Thank you for including a quantitative description of the measurement uncertainty.
- "Turbulence intensity is 1.5%" — These are very low turbulence levels, likely to arise offshore during stable atmospheric conditions. Since the experiment presumably did not apply a temperature gradient across the boundary layer, I think "close to laminar," as the authors state it, is a good description.. Also, given the confidence interval stated above, does this mean that the actual TI has a $\pm$ 27% confidence interval (1.5% $\pm$ 0.4% TI)?
- "reduced frequency:" — It may be helpful to readers to state that the reduced frequency is a normalized frequency similar to the Strouhal number.
- "Normalized mean velocity and turbulence intensity ($I$)" — The inflow average wind speed and TI are surprisingly variable across the rotor area. Does this arise from obstructions in the wind tunnel? Is this level of variability in $y$ and $z$ expected?
- "This process was performed in the frequency domain by computing the complex FFT, keeping the frequency components up to 3 Hz, and then utilizing the inverse FFT to reconstruct the signal in the time domain." — This sort of low-pass filtering can introduce convolution errors that are visible in the filtered time-domain data. Were any other techniques used (e.g., zero-padding, Butter or other optimal filter design) to reduce convolution errors?
- "12 platform motion cycles" — Is this a large enough sample size to describe spectra and turbulence moments given the non-stationary (periodic) nature of the flow statistics?
- "This proves that rotor load fluctuations are mainly due to variations in apparent wind speed caused by platform movement (Fontanella et al., 2021)." — Are the aerodynamic thrust and rotor torque in Figure 4 nominal curves based on the imposed platform motion and a constant wind speed or from measurements? How do the shown trends averaged over the 12 periods of motion? How do measured forces for each period compare to the nominal curves? It is not entirely clear here that fluctuations are due to apparent wind speed variations. How much of the loads are from acceleration of the platform itself?
- "A linear regression is fitted to the measurements and is included in Figure 5a and Figure 5b. This method of representation demonstrates that the loads change linearly with respect to the platform motion amplitude, as evidenced by the normalized points aligning with the regression line." —item "The velocity profile has a double-gaussian shape, and it is mostly unaffected by platform motion" — Can the authors elaborate why the velocity profile is asymmetric with y? There appears to be a lateral shift in the mean profile of approximately 0.15 m and the peak deficit in the right side of the plot shows complex behavior for all cases.
- "Figure 7c examines the velocity at $x_w = 3D$ and $y_i = 0.6$ m over one motion cycle" — Wouldn't it be more meaningful to show the velocity measurements and apply the sinusoidal fit to the collective data of all 12 periods of motion? If the velocity trends are truly periodic and the phase relationship is consistent, this should provide a better least-squares fit. Please also indicate fit quality.

- "As shown in the graph, the sine wave at the motion frequency does not fit the velocity data as well as it does in the 0.5 Hz and 1 Hz cases." — How much of the high frequency oscillations in Figure 7c arise from the rotor motion? A rotor speed of 240 rpm corresponds to 4 Hz, which is similar enough to the imposed platform motion that one could expect some interaction.
- "Among the conditions explored in this study, the strongest perturbation of the wake occurs when the reduced frequency of platform motion is 0.6" — This case is not shown in figure 7. Should it be?
- "We believe that coherent flow structures, larger for $f = 1$ Hz at 3D (Fig. 7), are dissipated as they move downstream by generating small-scale turbulence, which increases the turbulence intensity and accelerates the wake transition." — It is not clear how this conclusion is drawn from the data in Figure 9.

**Suggested References**

[1] Endre Tenggren et al. "A Numerical Study on the Effect of Wind Turbine Wake Meandering on the Power Production of Hywind Tampen". In: *Journal of Physics: Conference Series* 1669.1 (Oct. 2020), p. 012026. ISSN: 1742-6596. DOI: 10.1088/1742-6596/1669/1/012026. (Visited on 09/12/2024).
[2] Zhaobin Li, Guodan Dong, and Xiaolei Yang. "Onset of Wake Meandering for a Floating Offshore Wind Turbine under Side-to-Side Motion". In: *Journal of Fluid Mechanics* 934 (Mar. 2022), A29. ISSN: 0022-1120, 1469-7645. DOI: 10.1017/jfm.2021.1147. (Visited on 09/12/2024).

---

## Referee Comment (RC3)

Review of manuscript: wes-2024-140
Title: Wake Development in Floating Wind Turbines: New Insights and Open Dataset from Wind Tunnel Experiments
Authors: Fontanella et al

**Overall comments:**

The submitted paper performs a unique and comprehensive set of experiments of a scaled wind turbine in kinematic motion to investigate both unsteady loads and wakes relevant to floating turbines. The objective to simultaneously study loads and wakes is critically important and valuable to the wind energy science community. Generally speaking, the paper is well written and the experiments are well described and thorough.

There are aspects of the paper that can be improved to increase the value and impact of the experimental results. I hope the authors may consider the following comments in a revision.

**General comments:**

1. The results would be much easier to follow and to compare with existing literature (previous lab experiments, field measurements, and models) if they were reported nondimensionally. Dimensional results are specific to the setup and make it hard to compare to other studies.
2. Relatedly, the authors can more clearly explain and justify the choice of nondimensional parameters that were investigated here. How were the amplitudes and frequencies (Strouhal numbers) selected?
3. Several conclusions are oversimplified, for example:
   a. The level of agreement with the linear regression in Figure 5 is overstated, which is important, because it affects the degree to which a linear model is justified (it seems not to be justified).
   b. More problematically, in the conclusions on Line 350, the authors state: "The mean velocity within the wake, regardless of the direction of movement, closely matches that of the bottom-fixed scenario." This statement cannot be made, as it will depend on the nondimensional parameters. For some Strouhal numbers and amplitudes of motion, the wake will match bottom fixed, and for others not. The authors could rephrase as: "The mean velocity within the wake, regardless of the direction of movement, closely matches that of the bottom-fixed scenario **for the dynamic regimes we have investigated here**" or similar. Otherwise, the authors would need to investigate why their results differ from published studies including Messmer 2024 and others.
   c. Similar unclear points elsewhere in the study are identified below in point comments

**Point comments:**

1. The literature review is thorough and useful. It may be interesting to discuss the findings of recent articles on the subject of kinematic motion/unsteady inflow affects on power [1] and wakes [2,3], especially relevant to the studies of Messmer in 2024.

2. The motivation for the article on Line 47 is clear and concise. I would suggest also including research questions and hypotheses that you seek to address in this article.

3. Section 2.1: Presumably the previous studies referenced here included validation of the lab model against the DTU 10 MW reference, but that should be made more explicit here (and perhaps included in an Appendix) so that this article is self contained.

4. Section 2.1: The authors should report the nondimensional numbers that govern the dynamics explicitly for both the wind tunnel tests and also the true DTU 10 MW reference turbine. Specifically: Reynolds number, tip-speed ratio, and Mach number for 'fixed bottom' turbine operation.

5. Line 83: "Substantial agreement was found between the two measurements."
   This is a vague statement and the measurements are not shown. The measurements should be shown to confirm this or this statement should be removed

6. Line 85: It seems limiting to only have a single wire rather than a cross wire, especially if yaw misalignment and sway are investigated, where lateral velocities become non-negligible

7. Line 107: Relative to the literature review in the introduction, especially Messmer et al., 2024b, the experiments were selected to be performed in laminar inflow. More discussion of this choice would be appropriate, as it will reduce the relevance of the wake measurements to the true system.

8. Figure 2: There appears to be a consistent structure in the deviations from hub-height wind speed in figure 2(a). Do the authors have explanations for this? i.e. is it a boundary layer from the bottom surface? Interactions with the wall/flow speed up on the outside of the rotor. Boundary layer from the top surface?

9. Line 118: The experiments are conducted with a constant rotor speed. This is probably a good choice, because it eliminates controller feedback which would make the results more complicated. But I suggest discussing this choice and explaining it more.

10. Section 2.1: More discussion of the reduced frequencies investigated in this study compared to the full system is needed. Also, this is more commonly called the Strouhal number.

11. Line 138: Consider adding the angles introduced here to Figure 1

12. Line 148: "With surge, the blades velocity is equal to the platform velocity and the apparent wind experienced by the rotor is: …"
    The authors can more carefully define the 'apparent wind' terminology they are using because the velocity at the blades will be affected by induction. Similar comments throughout this section.

13. Section 5.1: Useful to report dimensional loads but it would be much more useful to also report the nondimensional results so it can be more directly compared to the full-scale system, other wind tunnels, and models. Likewise, plotting the results against reduced frequency would be better than dimensional frequency.

14. Line 217: "The differences in the mean value of Fx are likely due to the zero blade-pitch recalibration done during testing."
    Can the authors elaborate on what this means?

15. Figure 4: The experimental curves are remarkably smooth. Has any smoothing/postprocessing been applied to the plot?

16. Line 225: "This method of representation demonstrates that the loads change linearly with respect to the platform motion amplitude, as evidenced by the normalized points aligning with the

regression line."

I'm not sure I agree with this conclusion. It appears to me that the trend is not linear as evidenced by the lack of collapse at the higher frequencies.

17. Line 227: "Additionally, the loads exhibit a linear increase with frequency."
Same comment as above, but now with respect to frequency.

18. Line 233: "This deviation, already seen by Bergua et al. (2023), is attributed to fluctuations in rotor speed and the flexible response of the tower, which could affect the wind turbine behavior under these testing conditions."
Is there any evidence of this or is this speculation? Specifically, I thought the turbine was designed for rigidity as explained earlier. Are the authors sure there are not other physical mechanisms causing nonlinear response?

19. Line 238: Is there a reason the authors selected such a low yaw amplitude? Even fixed bottom turbines exhibit yaw variations of 10 degrees from turbulence.

20. Line 248: Similar to above, it's not clear to me that the thrust changes are linear.

21. Equation 8: Since the authors are using a single probe hot wire, this should be streamwise velocity rather than wind speed. Likewise for the streamwise standard deviation and streamwise TI.

22. Figure 7(c): Are these results phase averaged or instantaneous?

23. Line 275: "Among the conditions explored in this study, the strongest perturbation of the wake occurs when the reduced frequency of platform motion is 0.6."
Is this shown somewhere? If not, please add a result that proves this or remove such statements.

24. Line 282: "The asymmetry might be due to flow in inhomogeneity in the wind tunnel seen in Fig. 2, potentially reducing the correlation between wake flow structures. This hypothesis can be verified through numerical simulations of the experiment, incorporating a high-fidelity model of the wind tunnel inflow."
This is a pretty weak statement. Further  investigation should be performed with the available measurements.

25. Line 350: "The mean velocity within the wake, regardless of the direction of movement, closely matches that of the bottom-fixed scenario."
This statement cannot be made. This will depend on the parameters that govern the movement. One could say: for the parameters investigated, the mean velocity in the wake is similar to the fixed bottom scenario. But the statement as written is not correct.

26. Line 356: "In offshore environments turbulence intensity depends on the wave height and typically ranges from 3% to 6%."
This is a statement which is far too over-simplified. Turbulence intensity in offshore environments does not 'depend' on the wave height only, but may partially depend on the wave height, or better yet, is 'correlated' with the wave height.

**References**

[1] Wei, Nathaniel J., and John O. Dabiri. "Power-generation enhancements and upstream flow properties of turbines in unsteady inflow conditions." Journal of Fluid Mechanics 966 (2023): A30.

[2] Wei, Nathaniel J., Adnan El Makdah, JiaCheng Hu, Frieder Kaiser, David E. Rival, and John O. Dabiri. "Wake dynamics of wind turbines in unsteady streamwise flow conditions." Journal of Fluid Mechanics 1000 (2024): A66.

[3] Li, Zhaobin, Guodan Dong, and Xiaolei Yang. "Onset of wake meandering for a floating offshore wind turbine under side-to-side motion." Journal of Fluid Mechanics 934 (2022): A29.

---

## Author Response (AR1)

Politecnico di Milano
Department of Mechanical Engineering
Via La Masa 1, 20156, Milan
Italy

Wind Energy Science Discussion

Date: February 28, 2025
Subject: WES-2024-140 Final Response

Dear Referees,

We would like to express our sincere gratitude for the time and effort you have devoted to reviewing our manuscript and for providing such valuable feedback. Your comments have highlighted important aspects we had not fully addressed in the initial submission, and we believe that incorporating your suggestions has significantly improved the quality and impact of our work.

We have carefully considered each of your comments and have revised the manuscript accordingly.

On behalf of all Authors,
sincerely,

Alessandro Fontanella

**Response to Referees comments**

In this document, **R1** refers to Referee 1, **R2** to Referee 2, and **AC** to our response. We have organized the comments by section and, where possible, grouped similar comments together, providing a single response to avoid redundancy.

After the point-by-point response, we have included a copy of the revised article, where the main changes made to address the Referees' comments are highlighted in blue.

**General comments**
**R1**: One of the main considerations overlooked in the current work is potential interaction between periodic turbulence statistics in the wake, and the imposed platform motion. These interactions could lead to harmonics visible in velocity measurements, which are currently shown in terms of the phase of the imposed platform motion.
**AC**: we introduced a new figure (Fig. 10) that illustrates the evolution of turbulence and periodic wakes structures across the wake through the power spectral density of the velocity.

**R2**: The results would be much easier to follow and to compare with existing literature (previous lab experiments, field measurements, and models) if they were reported nondimensionally. Dimensional results are specific to the setup and make it hard to compare to other studies.
**AC**: we agree that reporting results in non-dimensional quantities enhances comparability with other studies. We clarified the dependency on reduced frequency in the figures showing loads and wake quantities. Additionally, we normalized velocities by the free stream wind speed and used diameters for the axes in wake figures.

**R2**: Relatedly, the authors can more clearly explain and justify the choice of nondimensional parameters that were investigated here. How were the amplitudes and frequencies (Strouhal numbers) selected?
**AC**: Thank you for your comment. We have added an explanation of how frequency and amplitude values were selected in Sect. 3.2 Platform motion conditions.

**R2**: The level of agreement with the linear regression in Figure 5 is overstated, which is important, because it affects the degree to which a linear model is justified (it seems not to be justified).
**AC**: We noted that "The variation in loads for motion cases at a frequency of 2 Hz exceeds the expected linear trend." This behavior was seen in past experiments but not simulations, suggesting that non-idealities in the experimental setup cause the higher-than-linear response at 2 Hz. These non-idealities should be considered when comparing experimental and simulation results but do not significantly impact the trends in wake dynamics.

**R2**: More problematically, in the conclusions on Line 350, the authors state: "The mean velocity within the wake, regardless of the direction of movement, closely matches that of the bottom-fixed scenario." This statement cannot be made, as it will depend on the nondimensional parameters. For some Strouhal numbers and amplitudes of motion, the wake will match bottom fixed, and for others not. The authors could rephrase as: "The mean velocity within the wake, regardless of the direction of movement, closely matches that of the bottom-fixed scenario for the dynamic regimes we have investigated here" or similar. Otherwise, the authors would need to investigate why their results differ from published studies including Messmer 2024 and others.

**AC**: We agree with your comment and have rephrased the sentence to clarify that our conclusion is valid for the conditions examined in the experiment.

**Abstract**
**R1**: "NETTUNO" — Acronym used before it is defined.
**AC**: NETTUNO refers to the project titled "Understanding Turbine-Wake Interaction in Floating Wind Farms Through Experiments and Multi-Fidelity Simulations," as noted in the Acknowledgments. For readability, we use the short title in the main text.

**1 Introduction**
**R1**: "pre-commercial stage" — Is this still true? There are commercial deployments of floating
offshore turbines already.
**AC**: It is uncertain whether floating wind has reached a commercial stage or if the current projects are still intended to demonstrate the technology's capabilities. To clarify, we rephrased the sentence to state that there are only a few small-sized projects.

**R1**: "Very recently, Ozinan et al. (2024) studied the near wake of a 2 MW floating wind turbine and found no evident impact of wave-induced motions on the average velocity of the wake, partially contrasting theoretical speculations." — Does this undercut the motivation for the work? Please clarify how findings in this reference influence the NETTUNO project.
**AC**: We clarified that full-scale experiments are impacted by uncertainties related to measurements and the lack of information regarding wind conditions and platform movements. Additionally, we explained that controlled wind tunnel experiments are essential for interpreting full-scale measurements and supporting their conclusions.

**R1**: "10 MW" and "15 MW turbine" — Please specify the model of turbine used as a basis of the scaled experiments.
**AC**: We specified the wind turbine models.

**R2**: The literature review is thorough and useful. It may be interesting to discuss the findings of recent articles on the subject of kinematic motion/unsteady inflow affects on power [1] and wakes [2,3], especially relevant to the studies of Messmer in 2024.
**AC**: We cited some of the articles suggested by you and the other referee that were useful to comment on our results and support our conclusions.

**R2**: The motivation for the article on Line 47 is clear and concise. I would suggest also including research questions and hypotheses that you seek to address in this article.
**AC**: We implemented your suggestion by adding the research questions and hypothesis in the introduction.

**2.1 Wind turbine**
**R1**: "The design of the blades and tower focused on maximizing stiffness to minimize their aeroelastic response." — Since measurement of structural loads are among the project goals, can the authors provide some more detail on how aeroelastic scaling was approached for the wind turbine model? The authors state that the goal is to maximize blade stiffness, but this is not likely to produce results that scale up to the 10 MW turbine model. More justification for this choice would help readers understand the approach.
**AC**: In Sect. 2.1, we clarified that the wind turbine scale model is designed to isolate aerodynamic loads and wake response due to platform motion from those caused by blade and tower flexibility, which are beyond this study scope.

**R2**: Section 2.1: Presumably the previous studies referenced here included validation of the lab model against the DTU 10 MW reference, but that should be made more explicit here (and perhaps included in an Appendix) so that this article is self contained.
**AC**: An Appendix has been included to present the validation of the aerodynamic design of the wind turbine scale model against its full-scale reference.

**R2**: Section 2.1: The authors should report the nondimensional numbers that govern the dynamics explicitly for both the wind tunnel tests and also the true DTU 10 MW reference turbine. Specifically: Reynolds number, tip-speed ratio, and Mach number for 'fixed bottom' turbine operation.
**AC**: The chord-based Reynolds number has been added in section 2.1. It is stated in section 2.1 that the tip-speed ratio of the scaled turbine is identical to that of the full-scale turbine, and section 3.1 specifies that the turbine operates at a TSR of 7.5. Discussing the Mach number seems unnecessary.

**R1**: "ensuring the rotor was vertical to the wind tunnel floor" — Does this mirror the 10 MW reference turbine design? Typically, the tower is nominally vertical but the rotor is not. Perhaps it would be instructive to state here that aligning the shaft with the mean flow is an intentional simplification, to limit loads to the axial thrust and moment in the baseline case.
**AC**: We implemented your suggestion at the end of Sect. 2.1.

**2.2 Measurements**
**R1**: "six-component loads" — Please be more specific about the loads measured in the introduction. This statement seems to indicate that only aggregate rotor forces and moments were measured. Were the tower bending or twist measured? Blade loads?
**AC**: We clarified that the test campaign measured the three forces and three moments at the interface between the tower-top and nacelle. Tower bending and blade loads were not measured because these components are rigid in the wind turbine scale model. Therefore, tower loads can be derived from tower-top loads, and blade loads can be determined from the thrust force measurement.

**R1**: "The wind velocity in the turbine wake was measured using a hot wire anemometer." — While a hotwire is certainly a good system for pointwise turbulence measurements, using a single probe in isolation makes coherence and spatial correlations impossible. With such a capable traverse system, why not make multiple measurements simultaneously?
**AC**: The study used a single hot-wire anemometer assuming periodic wake response to platform motion. Velocity measurements were phase-aligned with the platform motion signal for spatial correlation. Accurate turbulence correlation requires simultaneous multi-point measurements, achievable through hot-wire probe arrays or advanced techniques like Particle Image Velocimetry (PIV). This is detailed in Sect. 2.2.

**R2**: Line 85: It seems limiting to only have a single wire rather than a cross wire, especially if yaw misalignment and sway are investigated, where lateral velocities become non-negligible.
**AC**: As noted by the reviewer, certain motion conditions may cause oscillatory velocities in unmeasured components. Section 2.2 explains that we measured the velocity in the xi direction (u) because it is predominant in the studied wake region. The time-averaged values of the other two velocity components are expected to be significantly lower than u, so our focus remained on the xi direction. The decision to measure only the u component is consistent with other recent experimental studies on wind turbine wakes.

**R1**: "The resulting average extended uncertainties, at a 95% confidence level, are 0.17 m/s for flow speed and 0.4% for turbulence intensity." — Thank you for including a quantitative description of the measurement uncertainty.
**AC**: Thank you.

**R2**: Line 83: "Substantial agreement was found between the two measurements." This is a vague statement and the measurements are not shown. The measurements should be shown to confirm this or this statement should be removed
**AC**: At the start of Section 2.2, we have clarified the purpose of the two Pitot tubes and linked these instruments to the measurements described in the text.

**3 Test scenarios**
**R1**: "Turbulence intensity is 1.5%" — These are very low turbulence levels, likely to arise offshore during stable atmospheric conditions. Since the experiment presumably did not apply a temperature gradient across the boundary layer, I think "close to laminar," as the authors state it, is a good description.
**AC**: we agree with your comment, we left "close to laminar" and we removed the reference to the paper describing turbulence levels in offshore conditions.

**R2**: Line 107: Relative to the literature review in the introduction, especially Messmer et al., 2024b, the experiments were selected to be performed in laminar inflow. More discussion of this choice would be appropriate, as it will reduce the relevance of the wake measurements to the true system.
**AC**: Agreed. In Section 3, we explained that laminar inflow ensures a controlled environment, reducing external turbulence. This allows for a precise evaluation of the wake structure and turbulence generated by the turbine and platform motion.

**R1**: Also, given the confidence interval stated above, does this mean that the actual TI has a ± 27% confidence interval (1.5% ± 0.4% TI)?
**AC**: We explained in the text that the average turbulence intensity is 1.5% with variations ranging from a minimum of 1.2% to a maximum of 2.1% across the rotor disk.

**R1**: "reduced frequency:" — It may be helpful to readers to state that the reduced frequency is a normalized frequency similar to the Strouhal number.
**AC**: we added a note as you suggested.

**R2**: Section 2.1: More discussion of the reduced frequencies investigated in this study compared to the full system is needed. Also, this is more commonly called the Strouhal number.
**AC**: In section 3.1 we added an explanation of how the frequencies of the experiment were selected. In response to R1 comment, we explained that the reduced frequency is similar to the Strouhal number. We prefer to use reduced frequency because this is often used in studies on floating wind turbines.

**R1**: "Normalized mean velocity and turbulence intensity (I)" — The inflow average wind speed and TI are surprisingly variable across the rotor area. Does this arise from obstructions in the wind tunnel? Is this level of variability in y and z expected?
**R2**: Figure 2: There appears to be a consistent structure in the deviations from hub-height wind speed in figure 2(a). Do the authors have explanations for this? i.e. is it a boundary layer from the bottom surface? Interactions with the wall/flow speed up on the outside of the rotor. Boundary layer from the top surface?
**AC**: we clarified the spatial variability of mean wind speed and turbulence intensity arises from the wind tunnel design and is comparable to the one achieved in other wind tunnel studies on wind turbines.

**3.1 Wind conditions and wind turbine settings**

**R2**: Line 118: The experiments are conducted with a constant rotor speed. This is probably a good choice, because it eliminates controller feedback which would make the results more complicated. But I suggest discussing this choice and explaining it more.

**AC**: We explained that reproducing the actions of a feedback controller in scale model experiments is challenging and introduces uncertainty in the measurement of aerodynamic loads.

**3.2 Platform motion conditions**

**R2**: Line 138: Consider adding the angles introduced here to Figure 1.

**AC**: done.

**R2**: Line 148: "With surge, the blades velocity is equal to the platform velocity and the apparent wind experienced by the rotor is: ..." The authors can more carefully define the 'apparent wind' terminology they are using because the velocity at the blades will be affected by induction. Similar comments throughout this section.

**AC**: We agree with your comment and have revised the text.

**4 Data processing**

**R1**: "This process was performed in the frequency domain by computing the complex FFT, keeping the frequency components up to 3 Hz, and then utilizing the inverse FFT to reconstruct the signal in the time domain." — This sort of low-pass filtering can introduce convolution errors that are visible in the filtered time-domain data. Were any other techniques used (e.g., zero-padding, Butter or other optimal filter design) to reduce convolution errors?

**AC**: The filtering process based on FFT-IFFT was chosen over two other options with optimal filters because the latter were less accurate in isolating the aerodynamic loads response due to platform motion. This is shown below.

Figure 1 presents the spectra of aerodynamic thrust and torque obtained from the tower top load cell under conditions of pitch motion with a frequency of 1Hz and an amplitude of 1.3°. The spectra display peaks corresponding to the rotor frequency (1P) and its multiple harmonics (e.g., 2P, 3P, etc.). These peaks result from a mass imbalance of 10% in one of the blades, aerodynamic imbalance (such as a slight blade-pitch offset), and the tower shadow effect.

The goal of filtering was to remove the loads response due to rotor asymmetry. Three filtering options were compared:

- 4th-order Butterworth filter with a 3 Hz cut-off frequency, compensating for delay.
- Stop-band filters with bandpass of 3-11 Hz and 13-22 Hz, plus a Butterworth filter with a 27 Hz cut-off. Stop-band filters use a minimum order with 60 dB attenuation. Stop-band filters and the low-pass filter compensate for delay. The goal of this filtering was to remove the 2P, 4P harmonics, and frequencies above 6P.
- FFT-IFFT with a 3 Hz cut-off frequency.

[Figure]

**Figure 1**. Power spectral density (PSD) of aerodynamic thrust and torque in the scenario with platform pitch motion at a frequency of 1 Hz and an amplitude of 1.3° resulting from various filtering procedures. The platform motion frequencies (0.5, 1, and 2 Hz) considered in the experiment are marked by dotted lines.

Figure 2 presents the time series of the aerodynamic thrust and torque after applying filtering and phase-averaging techniques. The thrust force response obtained using Butterworth and FFT-IFFT filtering methods are similar, whereas the response from the Stop-band+low-pass filtering method shows oscillations at the 3P frequency of 12 Hz (i.e., 12 high-frequency cycles within one cycle of platform motion at 1Hz). The torque response filtered with Butterworth and Stop-band+low-pass methods retains some of the response at the 1P frequency of 4 Hz, which is not completely attenuated due to the gradual roll-in of the optimal filters.

[Figure]

**Figure 2**. Phase-averaged time series of aerodynamic thrust and torque in the scenario with platform pitch motion at a frequency of 1 Hz and an amplitude of 1.3° resulting from various filtering procedures.

Based on these observations, it was decided to discard the filtering based on optimal filters because they cannot eliminate harmonics at the 1P frequency without affecting the response at 2 Hz, which is where platform motion occurs. The FFT-IFFT filtering introduces convolution errors; however: 1) it removes the response at 1P without impacting the response at 2Hz, and 2) the convolution errors are smaller than the oscillations in the signals due to rotor asymmetry, which cannot be fully eliminated with optimal filters.

The same FFT-IFFT filtering process with a cut-off frequency of 3 Hz was also utilized in the OC6-Phase III project (https://doi.org/10.5194/wes-8-465-2023), which examined aerodynamic loads obtained in a wind tunnel experiment using a setup similar to the one used in the present study.

We updated the manuscript by adding a sentence in Sect. 4 to explain the selection of the filtering process.

**R1**: "12 platform motion cycles" — Is this a large enough sample size to describe spectra and turbulence moments given the non-stationary (periodic) nature of the flow statistics?
**AC**: A sensitivity analysis was conducted to examine the evolution of mean wind speed, turbulence intensity, and amplitude of the velocity spectrum at the frequency of platform motion. This is illustrated below for the scenario involving platform pitch motion with a frequency of 1 Hz and an amplitude of 1.3°. Figure 3 presents the progression of the three evaluated parameters as the number of platform motion cycles increases. Figure 4 depicts the development of the coefficient of determination (R2) of the three parameters in relation to their values obtained from evaluating 12 cycles of platform motion. In this context, $R^2$ measures how well a parameter computed on n cycles aligns with the parameter computed over 12 cycles. The results indicate that the three parameters exhibit minimal variation when the number of cycles exceeds 10. Among them, the mean wind speed converges the fastest, followed by turbulence intensity and spectrum amplitude. A sample size of 12 is considered a suitable balance between the accuracy of parameter estimation relevant to this study and the time required to conduct the measurements. A sentence was included at the end of Sect. 4 to summarize this analysis.

[Figure]

**Figure 3**. Progression of mean wind speed (left), turbulence intensity (middle), and amplitude of the velocity spectrum (right) at the frequency of platform motion as the number of platform motion cycles (nc) increases.

[Figure]

**Figure 4**. Evolution of the coefficient of determination of the three parameters of Figure 4 for increasing number of platform motion cycles.

**5.1 Aerodynamic rotor loads**

**R1.1**: By asking whether the traces in figure 4 represent "nominal curves," I mean to ask whether they are the measured responses of the platform or they represent a modeled or idealized response to the imposed motion.

**R2**: Figure 4: The experimental curves are remarkably smooth. Has any smoothing / postprocessing been applied to the plot?

**AC**: We clarified that the curves are derived from measurements, and the smooth time series indicate that the number of motion cycles included in the measurements, as well as the data processing method, effectively isolate the aerodynamic loads response to platform motion.

**R2**: Section 5.1: Useful to report dimensional loads but it would be much more useful to also report the nondimensional results so it can be more directly compared to the full-scale system, other wind tunnels, and models. Likewise, plotting the results against reduced frequency would be better than dimensional frequency.

**AC**: we agree that reporting results in non-dimensional quantities enhances comparability with other studies. We clarified the dependency on reduced frequency in the figures showing loads and wake quantities. Additionally, we normalized velocities by the free stream wind speed and used diameters for the axes in wake figures.

**R2**: Line 217: "The differences in the mean value of Fx are likely due to the zero blade-pitch recalibration done during testing." Can the authors elaborate on what this means?

**AC**: we added an explanation in Sect. 3.1. The test procedure required setting the zero blade-pitch position to ensure consistent operation of the wind turbine. This process was performed individually for each blade using an inclinometer and it was repeated regularly throughout the test campaign. The resulting zero blade-pitch position achieved an accuracy of ±0.75°.

**R1.1**: "A linear regression is fitted to the measurements and is included in Figure 5a and Figure 5b. This method of representation demonstrates that the loads change linearly with respect to the platform motion amplitude, as evidenced by the normalized points aligning with the regression line.: --- The loads response indicated by the markers in Figure 5 do not appear to follow a linear trend. For the 2 Hz case, both the thrust and the Torque seem to deviate from a linear trend established by the other cases.

**R2**: Line 225: "This method of representation demonstrates that the loads change linearly with respect to the platform motion amplitude, as evidenced by the normalized points aligning with the regression line." I'm not sure I agree with this conclusion. It appears to me that the trend is not linear as evidenced by the lack of collapse at the higher frequencies.

**R2**: Line 227: "Additionally, the loads exhibit a linear increase with frequency." Same comment as above, but now with respect to frequency.

**AC**: We specified that "the loads change linearly with respect to the platform motion amplitude" is valid for the frequencies of 0.5 Hz and 1 Hz. It was already said that "the variation in loads for motion cases at a frequency of 2 Hz exceeds the expected linear trend".

**R2**: Line 233: "This deviation, already seen by Bergua et al. (2023), is attributed to fluctuations in rotor speed and the flexible response of the tower, which could affect the wind turbine behavior under these testing conditions." Is there any evidence of this or is this speculation? Specifically, I thought the turbine was designed for rigidity as explained earlier. Are the authors sure there are not other physical mechanisms causing nonlinear response?

**AC**: We have explained that it is unlikely that the above-linear trend is due to other physical mechanisms. This conclusion was derived from high-fidelity simulations of the wind turbine involved in this experiment. Please inform us if you have an alternative explanation for the nonlinear response.

**R2**: Line 248: Similar to above, it's not clear to me that the thrust changes are linear.

**AC**: We changed the sentence saying that "pitch and surge movement leads to periodic thrust changes *driven* by apparent wind speed fluctuations" and avoiding the word "proportional" that can be mistaken for "linearly proportional".

**R2**: Line 238: Is there a reason the authors selected such a low yaw amplitude? Even fixed bottom turbines exhibit yaw variations of 10 degrees from turbulence.

**AC**: The referee's comment is unclear. While 10° can be the instantaneous inflow skew angle, dynamic yaw variations of 10° are rare. Additionally, it is challenging to correlate harmonic motion from our experiment to the stochastic motion experienced by floating wind turbines or turbulent inflow skew. Anyway, we have added details on the selection of yaw motion amplitudes in Sect. 3.2.

**5.2 Wake with platform pitch and surge motions**

**R2**: Equation 8: Since the authors are using a single probe hot wire, this should be streamwise velocity rather than wind speed. Likewise for the streamwise standard deviation and streamwise TI.

**AC**: We agree with your comment and have revised the text.

**R1.1**: "The velocity profile has a double-gaussian shape, and it is mostly unaffected by platform motion" --- Can the authors elaborate why the velocity profile is asymmetric with y? There appears to be a lateral shift in the mean profile of approximately 0.15 m and the peak deficit in the right side of the plot shows complex behavior for all cases.

**AC**: Thank you for this comment. Previous measurements of the wake of the same turbine also showed asymmetry. This asymmetry is explained by the interaction of the tower with the rotor, as investigated in the study "Wind tunnel investigation on the effect of the turbine tower on wind turbines wake symmetry." The text at the beginning of Sect. 5.2 has been revised to clarify the possible reason for the wake asymmetry.

**R1**: "Figure 7c examines the velocity at xw = 3D and yi = 0.6 m over one motion cycle" — Wouldn't it be more meaningful to show the velocity measurements and apply the sinusoidal fit to the collective data of all 12 periods of motion? If the velocity trends are truly periodic and the phase relationship is consistent, this should provide a better least-squares fit. Please also indicate fit quality.

**R2**: Figure 7(c): Are these results phase averaged or instantaneous?

**AC**: We have clarified that the results shown in Fig. 7c represent phase-averaged data. Additionally, we amended the captions of the remaining figures to emphasize that they display phase-averaged rather than instantaneous data. We also introduced the parameter $R^2$ in the figures to demonstrate the fit of the sine function to the velocity data points derived from phase averaging. This parameter is now used in the text to provide a more thorough description of the results.

**R1**: "As shown in the graph, the sine wave at the motion frequency does not fit the velocity data as well as it does in the 0.5 Hz and 1 Hz cases." — How much of the high frequency oscillations in Figure 7c arise from the rotor motion? A rotor speed of 240 rpm corresponds to 4 Hz, which is similar enough to the imposed platform motion that one could expect some interaction.

**AC**: In response to the referees' comments, we added a figure showing the power spectral density of wind speed at various positions in the turbine wake. The spectra show no flow response at 4 Hz.

**R1**: "Among the conditions explored in this study, the strongest perturbation of the wake occurs when the reduced frequency of platform motion is 0.6" — This case is not shown in figure 7. Should it be?

**R2**: Line 275: "Among the conditions explored in this study, the strongest perturbation of the wake occurs when the reduced frequency of platform motion is 0.6." Is this shown somewhere? If not, please add a result that proves this or remove such statements.

**AC**: The cases with a frequency of 1Hz have a reduced frequency fr = 1*2.4/4 = 0.6. To facilitate the readers, we include the reduced frequency value in the captions of all plots.

**R1**: "We believe that coherent flow structures, larger for f = 1 Hz at 3D (Fig. 7), are dissipated as they move downstream by generating small-scale turbulence, which increases the turbulence intensity and accelerates the wake transition." — It is not clear how this conclusion is drawn from the data in Figure 9.

**AC**: Thank you for your comment. We acknowledge that our comment to the results was not clear. To address this, we have introduced a new figure (Fig. 10) that illustrates the evolution of turbulence across the wake through the power spectral density of the velocity. This provides a clearer explanation of evolution of turbulence and flow structures coherent with platform motion.

**R2**: Line 282: "The asymmetry might be due to flow in inhomogeneity in the wind tunnel seen in Fig. 2, potentially reducing the correlation between wake flow structures. This hypothesis can be verified through numerical simulations of the experiment, incorporating a high-fidelity model of the wind tunnel inflow." This is a pretty weak statement. Further investigation should be performed with the available measurements.

**AC**: We agree that our comment was not justified properly. In response to R1 we explained that wake asymmetry can be the consequence of the interaction between the individual wakes of tower and rotor. Moreover, we introduced a new figure (Fig. 10) that illustrates the evolution of turbulence across the wake through the power spectral density of the velocity. With the new figure, we explain that dissipation of coherent flow structures occurs due to interaction with other turbulent structures and because of turbulent mixing. This process can be asymmetric because of the asymmetry of the flow (Fig.2) and the tower wake affecting the rotor wake.

**Conclusions**

**R2**: Line 350: "The mean velocity within the wake, regardless of the direction of movement, closely matches that of the bottom-fixed scenario." This statement cannot be made. This will depend on the parameters that govern the movement. One could say: for the parameters investigated, the mean velocity in the wake is similar to the fixed bottom scenario. But the statement as written is not correct.

**AC**: The conclusions were adjusted to state that, for the wind turbine operating conditions, motion scenarios, and distances from the rotor examined in this experiment, the mean velocity in the wake is comparable to that of the bottom-fixed case.

**R2**: Line 356: "In offshore environments turbulence intensity depends on the wave height and typically ranges from 3% to 6%." This is a statement which is far too over-simplified. Turbulence intensity in offshore environments does not 'depend' on the wave height only, but may partially depend on the wave height, or better yet, is 'correlated' with the wave height.

**AC**: The reference to wave height has been removed and the sentences in the conclusions have been rephrased.

Citing these articles is useful as they provide theoretical context about wake dynamics under along-wind and side-to-side motions, thereby strengthening the manuscript.

**Additional changes**

We have modified the colors of the lines in the figures after one of the authors noticed that the previous color scheme could be difficult to distinguish when printing the article.

The text has been thoroughly revised to improve readability and clarity.

[revised manuscript text omitted]

---

## Author Response (AR2)

Politecnico di Milano
Department of Mechanical Engineering
Via La Masa 1, 20156, Milan
Italy

Wind Energy Science Discussion

Date: April 22, 2025
Subject: WES-2024-140 – Manuscript needs minor revisions

Dear Editor and Referees,

Thank you for taking the time to review our manuscript for a second time. We have carefully addressed all comments from Referee 2 and have also taken this opportunity to thoroughly revise the manuscript to further improve its clarity.

On behalf of all Authors,
sincerely,

Alessandro Fontanella